

# Continuous CH₄ and $\delta^{13}CH_4$ measurements in London demonstrate under-reported natural gas leakage

Eric Saboya[1,2], Giulia Zazzeri[1], Heather Graven[1,2], Alistair J. Manning[3], and Sylvia Englund Michel[4]

[1]Department of Physics, Imperial College London, London, UK
[2]Grantham Institute – Climate Change and the Environment. Imperial College London, London, UK
[3]UK Met Office, Exeter, EX1 3PB, UK
[4]Institute of Artic and Alpine Research, University of Colorado, Boulder, CO, USA

*Correspondence to*: Eric Saboya (ess17@ic.ac.uk) and Heather Graven (h.graven@imperial.ac.uk)

**Abstract.** Assessment of bottom-up greenhouse gas emissions estimates through independent methods is needed to
demonstrate whether reported values are accurate or if bottom-up methodologies need to be refined. We report atmospheric methane ($CH_4$) mole fractions and $\delta^{13}CH_4$ measurements from Imperial College London since early 2018 using a Picarro G2201-i analyser. Measurements from March 2018 to October 2020 were compared to simulations of $CH_4$ mole fractions and $\delta^{13}CH_4$ produced using the NAME dispersion model coupled with the UK National Atmospheric Emissions Inventory, UK NAEI, and the global inventory, EDGAR, with model spatial resolutions of ~2 km, ~10 km, and ~25 km. Observed mole
fractions were underestimated by 30-35 % in the NAEI simulations. In contrast, a good correspondence between observations and EDGAR simulations was seen. There was no correlation between the measured and simulated $\delta^{13}CH_4$ values for either NAEI or EDGAR, however, suggesting the inventories' sectoral attributions are incorrect. On average, natural gas sources accounted for 20-28 % of the above background $CH_4$ in the NAEI simulations, and only 6-9 % in the EDGAR simulations. In contrast, nearly 84 % of isotopic source values calculated by Keeling plot analysis (using measurement data from the afternoon)
of individual pollution events were higher than -45 ‰, suggesting the primary $CH_4$ sources in London are actually natural gas leaks. The simulation-observation comparison of $CH_4$ mole fractions suggests that total emissions in London are much higher than the NAEI estimate (0.04 Tg $CH_4$ yr⁻¹) but close to, or slightly lower than the EDGAR estimate (0.10 Tg $CH_4$ yr⁻¹). However, the simulation-observation comparison of $\delta^{13}CH_4$ and the Keeling plot results indicate that emissions due to natural gas leaks in London are being underestimated in both the UK NAEI and EDGAR.

## 1 Introduction

Urban areas are hotspots of greenhouse gas (GHG) emissions accounting for 70 % of anthropogenic GHG emissions (IPCC, 2014), making them important targets for GHG emissions mitigation (Duren and Miller, 2012; Hopkins et al., 2016). Urban areas account for 21 % of global anthropogenic $CH_4$ emissions (Marcotullio et al., 2013), and over 40 % of global $CH_4$ emissions from the waste, energy and transport sectors come from cities (Marcotullio et al., 2013).



In the UK, the National Atmospheric Emissions Inventory (NAEI) uses a bottom-up methodology to estimate $CH_4$ emissions and their spatial and sectoral distributions. The London region enclosed within the London orbital motorway comprise 0.65 % of the UK's land area yet accounts for 2.7 % of the UK's annual $CH_4$ emissions, and 9.1 % of the UK's annual fugitive (e.g. leaks from the natural gas distribution network) $CH_4$ emissions (NAEI, 2017). In the 2017 UK NAEI estimates, $CH_4$ from the

waste sector is the dominant source in London accounting for 52 % of London's $CH_4$ emissions, with fossil-fuel sources of methane (e.g. fugitive gas emissions) making up 41 % of London's $CH_4$ emissions (NAEI, 2017).

Attributing emissions to specific sources can be challenging when $CH_4$ sources are collocated. Isotopic measurements of $^{13}C/^{12}C$ in $CH_4$ ($\delta^{13}CH_4$) have become an established means for discriminating between sources of $CH_4$ (e.g. Fisher et al., 2017;

France et al., 2016; Tans, 1997). Sources can be distinguished by their different isotopic source signatures (e.g. Sherwood et al., 2017). UK isotopic signatures of waste have an average value of -58 ‰ whereas the average for natural gas is -36 ‰ (Zazzeri et al., 2017). The isotopic signatures of some sources have been found to exhibit spatiotemporal variations (Feinberg et al., 2018) so it is preferable to use regional values, when available, for interpreting atmospheric $\delta^{13}CH_4$ measurements (Feinberg et al., 2018; Hoheisel et al., 2019; Zazzeri et al., 2017).


Bottom-up $CH_4$ inventories tend to underestimate emissions in comparison to atmospheric measurements in urban regions (Brandt et al., 2014), including in London. Helfter et al. (2016) conducted eddy-covariance measurements from the BT Tower in central London between 2012-2014 and found emissions ($72 \pm 3$ ton $km^{-2}$ $yr^{-1}$) were more than double the NAEI inventory values, which was attributed to gas leaks being underestimated in the inventory (Helfter et al., 2016). Zazzeri et al. (2017)

also concluded that gas leaks were underestimated after finding many large gas leaks in mobile measurement surveys. However, a study using aircraft measurements from a single flight around the London region in 2016 suggested the UK NAEI was overestimating $CH_4$ emissions and they needed to be scaled down by 0.71 (0.66-0.79) to be consistent with the aircraft measurements on this particular day (Pitt et al., 2019).

Discrepancies between atmospheric measurements and bottom-up estimates have similarly been found in other urban regions. $CH_4$ observations in Boston, USA found natural gas emissions were 2-3 times higher than the emissions estimates from a customised inventory made up of local data (McKain et al., 2015). Airborne $CH_4$ flux measurements were more than double the 2015 EDGAR v5.0 inventory estimates in Berlin (Klausner et al., 2020). In Paris, Xueref-Remy et al. (2020) conducted mobile surveys for $CH_4$ and $\delta^{13}CH_4$ over 2012-2015 and found that emissions from the waste water treatment sector were

being underestimated in the AIRPARIF 2013 inventories.

Instruments capable of making continuous measurements of atmospheric $\delta^{13}CH_4$ have recently become available, yet only a few studies have deployed them to attribute $CH_4$ emissions in areas of mixed sources. Venturi et al. (2020) measured $\delta^{13}CH_4$





in Florence, Italy, over a few months in 2017 and found that $CH_4$ emissions in the city were mainly due to natural gas emissions.

In Cabauw, Netherlands Röckmann et al. (2016) deployed a dual isotope mass spectrometric system and a quantum cascade laser spectrometer to measure $\delta^{13}CH_4$. Model-data comparisons of $\delta^{13}CH_4$ across five months found simulations using the EDGAR inventory overestimated fossil-fuel $CH_4$ sources for this region. Assan et al. (2018) used a Picarro G2201-i to measure $\delta^{13}CH_4$, along with other atmospheric tracers, near a natural gas compressor station and found local sources were dominated by natural gas $CH_4$, with traffic-related and ruminant sources also present. The first network of continuous atmospheric $\delta^{13}CH_4$

measurements, using cavity ring-down spectroscopy (CRDS), comprised of four tall towers in the Marcellus Shale gas region, Pennsylvania (Miles et al., 2018) showed mean differences between flask and in situ $\delta^{13}CH_4$ were between 0.02 ‰ and 0.08 ‰, demonstrating CRDS has the capacity to make high-precision $\delta^{13}CH_4$ measurements that align with flask measurements.

Here we present over two years of continuous measurements of $CH_4$ mole fractions and $\delta^{13}CH_4$ values made from the South

Kensington campus of Imperial College London (ICL), in central London; the longest in situ $\delta^{13}CH_4$ measurement campaign reported to date. An automated Keeling plot analysis was created to determine the isotopic source values ($\delta_s$) of individual pollution events. We compare observations with atmospheric transport model simulations using 2017 UK NAEI and Emissions Database for Global Atmospheric Research (EDGAR) 2012 v4.3.2 (http://edgar.jrc.ec.europa.eu/overview.php; Janssens-Maenhout et al., 2012) bottom-up inventory estimates and their source apportionment for the London region. We used the UK

Met Office's Numerical Atmospheric-dispersion Modelling Environment (NAME v7.2; Jones et al., 2007) to transport these emissions under three different spatial resolutions to simulate the excess mole fractions and $\delta^{13}CH_4$ at ICL.

## 2 Methods

### 2.1 Measurements and site description

Measurements of $CH_4$ mole fractions and $\delta^{13}CH_4$ values were made at ICL using a Picarro G2201-i isotopic analyser beginning

in early 2018. Ambient air is sampled from an inlet mounted on a 2 m mast located on the southeast corner of the Huxley building roof (~26 m.a.g.l., 51.4999° N, 0.1749° W; Fig. 1). Measurement data are averaged into 1, 5, 20, and 60-minute intervals by GCWerks software (http://www.gcwerks.com). There are gaps in the data at times when the instrument was being used for laboratory tests. The mast is equipped with a weather station (ClimeMet) measuring 5-minute averaged wind speed and direction as well as atmospheric pressure and temperature. The air inlet is approximately level with the surrounding

rooftops and there are four main roads nearby.



**Figure 1: Map of the surrounding area of Imperial College London with the UK CH₄ 1 km NAEI estimates overlaid. The locations of large CH₄ sources are indicated. © OpenStreetMap contributors 2019. Distributed under the Open Data Commons Open Database License (ODbL) v1.0.**

There are several large potential sources of CH₄ in the vicinity of ICL that may influence the atmospheric CH₄ and $\delta^{13}$CH₄ measurements. The locations of some of these sources are highlighted on Fig. 1 with the UK NAEI CH₄ 1 km emissions superimposed. There are ~20 small sewage pumping stations and a waste facility south of the site in the Battersea area. An on-campus natural gas-fired power station is located in the basement of the Electrical and Electronic Engineering building (~200 m east of the inlet) with the stack emitting at ~ 52 m.a.g.l. (Sparks and Toumi, 2010). Eddy-covariance measurements of CO₂

previously conducted from the top of the adjacent building frequently detected emissions from the power station, and found a mean CO₂ flux of 18.6 µmol m⁻² s⁻¹ from the power station (Sparks and Toumi, 2010). This was ~70 % smaller than the UK CO₂ NAEI estimate of emissions from the power station at the time. The UK NAEI inventory estimates CH₄ emissions from the power station are $3.47 \times 10^3$ kg CH₄ yr⁻¹ (NAEI, 2017; Fig. 1).



## 2.2 Picarro calibrations and data correction

### 2.2.1 Measurement setup

Outside air is drawn into the lab through a 3/8" Synflex tube by a 30 litres-per-minute (lpm) KNF Laboport pump. Air is dried to water levels of 0.01 % using a Nafion Perma Pure gas dryer (PD-50-24) in the split sample configuration, with a 5 lpm diaphragm pump for the counterflow. The Nafion dryer was installed in August 2019. A water correction (Sect. 2.2.4) was applied to the sample air between March 2018 and August 2019 when the air was not dried. A Picarro 16-port manifold is used to switch valves and direct either outside air or standard tank air into the Picarro. A pressure controller between the manifold and the Picarro inlet (PC-100PSIA-D/5P, Alicat Scientific, Inc.) is used to keep the inlet pressure constant at approximately 14 psia.

### 2.2.2 Allan variance

An Allan variance (Allan, 1966) was calculated to measure the noise and drift response of the instrumentation over different averaging times. Two air tanks with ambient $CH_4$ mole fractions and $\delta^{13}CH_4$, referred to as the "low" standard (1900 ppb, -48.0 ‰) and "high" standard (2200 ppb, -47.0 ‰), have each been measured continuously for 24 h. An averaging time of four minutes has Allan variances of 0.3 ‰ and 0.2 ‰ for the low and high standard $\delta^{13}CH_4$ values (Fig. S1), respectively. This is consistent with previous tests carried out with Picarro G2201-i instruments (Miles et al., 2018; Rella et al., 2015). An averaging time of 20 minutes reduces the Allan variance to less than 0.1 ‰.

### 2.2.3 Calibration procedure and measurement uncertainty

Different calibration procedures were tested using one air tank as a working standard to correct for instrument drift and another air tank as a target tank to assess the standard deviation of the measurements. We assumed the response of the instrument was linear within the observed range (-50 to -42 ‰, 1900 to 4000 ppb) (Rella et al., 2015) and the working standard is stable and applied a one-point calibration by measuring the working standard once per day for an hour. The "bracketing technique" was used to correct for instrumental drift; i.e. the measurements were calibrated against the time-interpolated value of two adjacent standard measurements. There was an average daily drift of 0.25 ppb for $CH_4$ and 0.7 ‰ for $\delta^{13}CH_4$. Both air tanks were calibrated against two primary standards which were prepared at the Max Planck Institute for Biogeochemistry (MPI-BGC).

We tested calibrations based on the ratio or the difference between the measured value of the standard and the assigned calibrated value. Ratio-based calibration adjusts the slope, thus the correction varies with the measured value, whereas difference-based calibration adjusts the intercept and the correction does not vary across the measured value. Some studies recommend calibration of individual isotopologues (Griffith, 2018), while others use $\delta^{13}CH_4$ (Rella et al., 2015). The following calibration procedures for $\delta^{13}CH_4$ were tested:



1. $^{13}CH_4$ and $^{12}CH_4$ mole fractions were calibrated independently based on the *ratio* and then a calibrated $\delta^{13}CH_4$ was computed.

2. $^{13}CH_4$ and $^{12}CH_4$ mole fractions were calibrated independently based on the *difference* and then a calibrated $\delta^{13}CH_4$ was computed.

3. $\delta^{13}CH_4$ values were calibrated directly based on the *ratio*.

4. $\delta^{13}CH_4$ values were calibrated directly based on the *difference*.

We applied the different calibration procedures to 20-minute averaged measurements of the target from May 2019 to November 2019. All the calibration procedures performed comparably and reduced the standard deviation of the target tank $\delta^{13}CH_4$ values from 1.1 ‰ to 0.2 ‰. We chose to apply a one-point calibration based on the ratio between the measured standard value and the assigned $\delta^{13}CH_4$ value, which is the default calibration procedure used by GCWerks software. Rella et al. (2015) also applied calibration constants on the $\delta^{13}CH_4$ values rather than on the $^{13}CH_4$ values. The total $CH_4$ mole fraction was calculated using calibrated $^{12}CH_4$ and $\delta^{13}CH_4$ values, where $^{12}CH_4$ was also calibrated using a one-point calibration based on the ratio of the measured and assigned values. We regard the standard deviation of calibrated $CH_4$ mole fractions and $\delta^{13}CH_4$ in the target tank to be the best indicator of our measurement uncertainty, at 0.28 ppb and 0.2 ‰ for 20-minute averages after May 2019, and 1.8 ppb and 0.6 ‰ before May 2019. The mean of the standard deviations of each standard tank is 0.18 ppb and 0.5 ‰ before May 2019, and 0.16 ppb and 0.4 ‰ after May 2019, for $CH_4$ and $\delta^{13}CH_4$ respectively. The larger uncertainty before May 2019 is likely related to unexplained larger variations in the measurements of one of the reference tanks.

A correlation between atmospheric pressure and $\delta^{13}CH_4$ is seen in the raw measurements, which has been observed for CO in other Picarro analysers (Yver Kwok et al., 2015). The daily working tank calibrations removed the effect of atmospheric pressure variations over more than one day. For some days when atmospheric pressure changed rapidly within one day, artefacts appeared in $\delta^{13}CH_4$. The $\delta^{13}CH_4$ measurements were inspected for periods of high variability in atmospheric pressure and manually flagged to remove these artefacts.

Here, measurements at ICL were compared to the $\delta^{13}CH_4$ observations at the Mace Head Observatory carried out by the Institute of Arctic and Alpine Research (INSTAAR) of the University of Colorado. Therefore, we applied a value of +0.28 ‰ to correct for the laboratory offset between INSTAAR and MPI-BGC measurements (Umezawa et al., 2017).

### 2.2.4 Water correction

A cross interference from water has been observed on the $\delta^{13}CH_4$ values during the period March 2018-August 2019 when sample air was not dried. Rella et al. (2015) state the gas stream should be dried to <0.1 % water vapour content to increase measurement accuracy. Data measured before applying the Nafion dryer were corrected for the water vapour influence. To determine the correction coefficients, the water vapour concentration of a working standard with a $\delta^{13}CH_4$ value of -48.5 ‰



was varied using the setup in Fig. S2. A water correction range between 0 % and 2.2 % was generated by using two mass flow controllers to adjust the flow rate through the bubbler (Fig. S2). Five measurement cycles (each cycle being ~ 6 h) with $\delta^{13}CH_4$ values increasing with water vapour concentration are shown in Fig. S3a. The correction coefficients were determined by applying a least squares regression on the ratio of wet-to-dry $\delta^{13}CH_4$ values against the water concentration (Fig. S3b). Using the calibrated working standard $\delta^{13}CH_4$ value of -48.5 ‰ as the dry value, we calculated the following equation to correct for the water dependency:

$$dry\ data = \frac{observed\ data}{-0.0109\ X_{H2O} + 1.0023}. \qquad (1)$$

The errors of the linear regression parameters from the water vapour correction experiment were ~$10^{-3}$ ‰ suggesting there is no additional uncertainty resulting from the water vapour correction.

## 2.3 Keeling plot analysis

The Keeling plot technique (Keeling, 1961; Pataki et al., 2003) was used to assess isotopic signatures ($\delta_s$) of local and regional sources by analysing data across three different moving time intervals, or "windows" that were 12 h, 3 days, and 7 days in length. We expect that the $\delta_s$ values obtained with the 12 h window emphasize sources local to the measurement site, particularly the local emissions that accumulate in the nocturnal boundary layer. For the 3-day and 7-day time windows we used only daytime data between 13:00-17:00 when the planetary boundary layer (PBL) is at its largest to find $\delta_s$ values more representative of sources from the wider area. For all three time windows an orthogonal distance regression was applied to the 20-minute averaged data using an automated algorithm, similar to Röckmann et al. (2016). To ensure a coherent pollution event was captured, the $\delta_s$ value from each moving window was retained if the mole fractions varied by more than 150 ppb. The choice of this criterion (i.e. the mole fraction peak strength) was based on simulation experiments using pseudo data (Supplementary material).

### 2.4 Atmospheric simulations

### 2.4.1 NAME footprints

Simulations of atmospheric $CH_4$ at ICL were performed using the UK Met Office Lagrangian dispersion model NAME with meteorological fields from the UK Met Office's Unified Model (UM). NAME back-trajectories were used to calculate "footprints" of surface emission sensitivities. Each grid cell of the footprint describes the impact an emission from that grid cell would have on the mole fraction measured at the receptor site at a certain time (Manning et al., 2011; Rigby et al., 2012).

Three sets of hourly footprints were generated, each with a different horizontal spatial resolution: ~25 km, ~10 km, and ~2 km (Table 1). The domain of the 2 km resolution footprints covers the British Isles and a small portion of northern Europe, the domain of the 10 km resolution footprints covers most of Europe, and the domain of the 25 km resolution footprints extends





to central Northern America (Fig. 2). The 2 km and 10 km simulations used a 6 day back-trajectory duration whereas the 25 km simulations used a 30 day back-trajectory duration. Particle release rates of $2 \times 10^4$ h$^{-1}$ were used for the 25 km and 10 km footprints and $1.5 \times 10^4$ h$^{-1}$ for the 2 km footprints. Footprints used the Met Office UM $0.0135^o \times 0.0135^o$ UKV meteorological fields over the UK and UM $0.1406^o \times 0.0938^o$ global meteorological fields for the rest of the modelling domain. To compare simulations that used footprints with different modelling domains we created nested footprints that used the higher resolution footprints for the inner domain and the coarser footprints for the outer domain(s); Table 2.

**Table 1: NAME model parameters used for each of footprints.**

| Footprint | Horizontal spatial resolution | Particle release rate | Back-trajectory duration |
|---|---|---|---|
| 25 km | $\mathbf{0.352^o \times 0.234^o}$ | $\mathbf{20000}$ h$^{-1}$ | 30 days |
| 10 km | $\mathbf{0.10^o \times 0.10^o}$ | $\mathbf{20000}$ h$^{-1}$ | 6 days |
| 2 km | $\mathbf{0.020^o \times 0.020^o}$ | $\mathbf{15000}$ h$^{-1}$ | 6 days |

**Table 2: Summary of atmospheric CH$_4$ simulations. WetCHARTs and GFED4 were used for wetland and biomass burning emissions in all simulations.**

| Simulation | Footprints | Anthropogenic emissions |
|---|---|---|
| EDGAR-25km | 25 km | EDGAR |
| EDGAR-10km | 10 km nested in 25 km | EDGAR |
| NAEI-25km | 25 km | NAEI in UK, EDGAR outside UK |
| NAEI-2km | 2 km nested in 10 km nested in 25 km | NAEI in UK, EDGAR outside UK |

Footprints were combined with gridded emissions (Sect. 2.4.2) to simulate CH$_4$ mole fractions above the background mole fractions outside the footprint domain (i.e. excess CH$_4$ mole fractions). To compare the simulated excess CH$_4$ mole fractions to the measurements at ICL, we subtract daily background CH$_4$ mole fractions from the Mace Head Observatory (Arnold et al., 2018; Manning et al., 2011) from the 20-minute averaged measurements at ICL.

Simulated atmospheric $\delta^{13}$CH$_4$ ($\delta_{air}$) were calculated from a weighted average of the isotopic signatures of individual source sector components of excess CH$_4$ using the NAME simulations, and the background $\delta^{13}$CH$_4$ ($\delta_{bg}$) at Mace Head following:

$$\delta_{air} = \frac{\delta_{bg}C_{bg} + \sum_i \delta_i C_i}{C_{bg} + \sum_i C_i}. \quad (2)$$

Where $C_i$ and $\delta_i$ are the excess CH$_4$ and isotopic signatures of the individual source sectors, and $C_{bg}$ and $\delta_{bg}$ are the background CH$_4$ mole fraction and $\delta^{13}$CH$_4$.





Background $\delta^{13}CH_4$ values were calculated using measurements at Mace Head by following the method outlined in Manning
et al. (2011). Footprints at Mace Head are used to assess which measurements were not influenced by significant emissions
and are suitable as background measurements. We fit a curve of multiple harmonics (e.g. Jones et al., 2015) to the background
measurements at Mace Head from January 2018 to May 2020. We extrapolate to October 2020 by fitting a linear trend to the
data and assuming the same seasonal cycle to obtain a time series of daily $\delta^{13}CH_4$ values that match the period of ICL
observations.

Table 3 lists the isotopic signature assigned to each source sector in the UK NAEI and EDGAR inventories, based on the UK-
specific isotopic source signatures from Zazzeri et al. (2017). For anthropogenic source sectors that did not have a UK-specific
isotopic source signature (petroleum refining, 1A1b, and Oil, 1B2a, in EDGAR) global values from Sherwood et al. (2017)
were used. Some source sectors are composed of multiple sources with different isotopic source signatures, for example the
waste sector includes landfill sites and waste water treatment facilities. In this case the weighted average of the different
sources, based on the UK emissions reported to the UNFCCC (https://di.unfccc.int/comparison_by_category), were used to
calculate the isotopic source signature of that source sector.





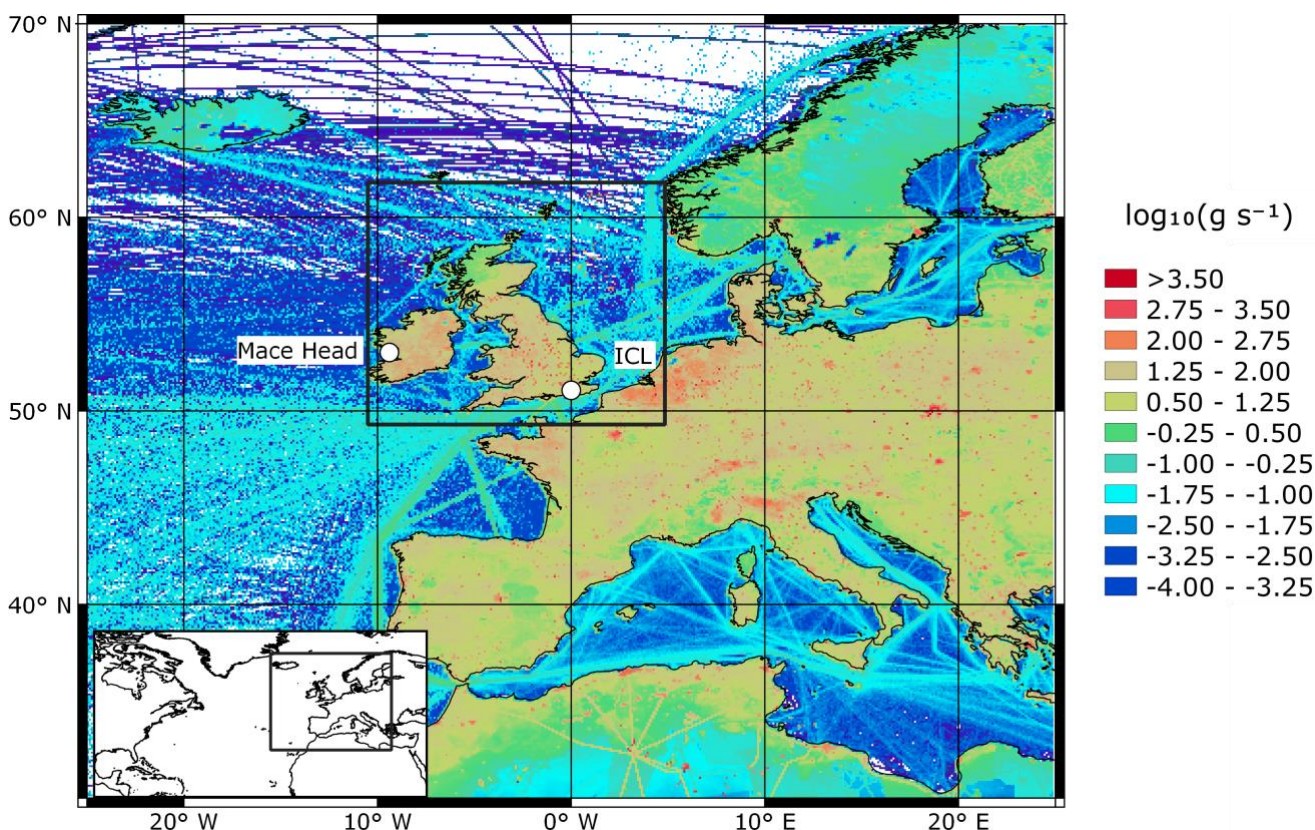

**Figure 2: The NAME footprint modelling domains. The inset map denotes the area encompassed by the 25 km footprints. The black box denotes the domain of the 10 km footprints, which is shown in the main frame along with the $0.1° \times 0.1°$ EDGAR emissions. The black box surrounding the British Isles denotes the 2 km footprint domain.**





**Table 3: The correspondence and allocation of methane sources between NAEI and EDGAR along with the assigned δ¹³CH₄ value for each source sector.**

| Source sector | UK NAEI SNAP sector | EDGAR v4.3.2 IPCC 1996 specification sector | Assigned $\delta^{13}CH_4$ $\pm 1\sigma$ (‰) | $\delta^{13}CH_4$ reference |
|---|---|---|---|---|
| Combustion in energy production and transfer | SNAP 01 | 1A1a | $-25 \pm 3$ | Zazzeri et al. (2017) |
| Non-industrial combustion | SNAP 02 | 1A4 | $-25 \pm 3$ | Zazzeri et al. (2017) |
| Combustion in industry | SNAP 03 | 1A2 | $-25 \pm 3$ | Zazzeri et al. (2017) |
| Production processes | SNAP 04 | 2B, 2C1a, 2C1c, 2C1d, 2C1e, 2C1f, 2C2 | $-25 \pm 3$ | Zazzeri et al. (2017) |
| Extraction and distribution of fossil fuels | SNAP 05 | 1A1b, 1A1c, 1A5b1, 1B1a 1B1b, 1B2a, 1B2b5, 1B2c, 2C1b | $-37 \pm 3$ | Sherwood et al. (2017); Zazzeri et al. (2017) |
| Road transport | SNAP 07 | 1A3b | $-20 \pm 3$ | Zazzeri et al. (2017) |
| Other transport | SNAP 08 | 1A3a, 1A3c, 1A3d, 1A3e, 1C2 | $-20 \pm 3$ | Zazzeri et al. (2017) |
| Waste treatment and disposal | SNAP 09 | 6A, 6B, 6C, 6D, | $-57 \pm 3$ | Zazzeri et al. (2017) |
| Agriculture | SNAP 10 | 4A, 4B, 4C, 4D | $-64 \pm 3$ | Zazzeri et al. (2017) |
| Wetlands (WetCHARTs) | | | $-71 \pm 1$ | Fisher et al. (2017) |
| Biomass burning (GFED4) | | | $-28 \pm 3$ | Zazzeri et al. (2017) |



### 2.4.2 Emissions data

We used two sources of anthropogenic $CH_4$ emissions data. The first is the Emissions Database for Global Atmospheric Research (EDGAR) v4.3.2 for the year 2012 with $0.1^o \times 0.1^o$ spatial resolution. The second is the UK National Atmospheric Emissions Inventory (NAEI) for 2017 with 1 km × 1 km spatial resolution, where we added point source emissions to the mapped emissions (which omit point sources) using the locations of the point sources. The NAEI is only available for the UK, so for simulations using the NAEI we created a hybrid emissions map with NAEI emissions for the UK and EDGAR emissions for outside the UK. Both emissions inventories have a yearly time resolution but neither provide gridded numerical uncertainties.

The two inventories use different sectoral definitions. The UK NAEI uses CORINAIR Selected Nomenclature for sources of Air Pollution (SNAP) in which sources are allocated to one of 11 categories, whereas EDGAR follows the 1996 IPCC source sector specification. Table 3 shows how we aligned the sources between inventories.

For wetland emissions we used the mean of the 2015 extended ensemble WetCHARTs inventory (Bloom et al., 2017). The extended ensemble consists of 18 models with a spatial resolution of $0.5^o \times 0.5^o$ and a monthly temporal resolution. For biomass burning emissions we used the Global Fire Emissions Database, v4 (GFED4; Van Der Werf et al., 2017) for 2016 at $0.25^o \times 0.25^o$ resolution and a monthly temporal resolution. To avoid double counting we excluded agricultural waste burning emissions from GFED4.

The four sets of anthropogenic emissions for the London area are shown in Fig. 3a-d. The UK NAEI emissions are approximately 2.5 times smaller than the EDGAR emissions for the London area (Table 4; Fig. 3e), but 8 % smaller than the EDGAR emissions across the UK (Fig. 3f). The 2 km NAEI and 10 km EDGAR show high emissions from individual grid cells that are smoothed out in the coarser 25 km EDGAR grid (Fig. 3a) and 25 km NAEI grid (Fig. 3c). Subtracting the 25 km NAEI emissions from the 25 km EDGAR emissions (Fig. 3e-f) indicates the largest differences between inventories were in cities; London, Birmingham and the Leeds-Sheffield area, which have higher emissions in the EDGAR inventory.

**Table 4: EDGAR and NAEI emissions for the UK and London. $\delta^{13}CH_4$ is the weighted average of different emission sectors using isotopic source signatures in Table 3.**

| Region | EDGAR emissions (Tg $CH_4$ $yr^{-1}$) | NAEI emissions (Tg $CH_4$ $yr^{-1}$) | EDGAR $\delta^{13}CH_4$ signature (‰) | NAEI $\delta^{13}CH_4$ signature (‰) |
|---|---|---|---|---|
| UK | 2.25 | 2.08 | -51.7 | -30.5 |
| London | 0.10 | 0.04 | -53.7 | -47.7 |





We considered four combinations of footprints coupled with anthropogenic emissions data: (i) the 25 km footprints combined with the EDGAR emissions (EDGAR-25km); (ii) the 10 km footprints nested in the 25 km footprints combined with the EDGAR emissions (EDGAR-10km); (iii) the 25 km footprints combined with the UK NAEI emissions for the UK and the EDGAR emissions for the rest of the domain (NAEI-25km); and (iv) the 2 km footprints nested in the 10 km and 25 km footprints combined with the UK NAEI emissions for the UK and EDGAR for the rest of the domain (NAEI-2km). These are

summarised in Table 2.





**Figure 3: London CH4 emissions from (a) EDGAR v4.3.2 (2012) scaled at 0.352º × 0.234º , (b) EDGAR scaled at 0.10º × 0.10º, (c) UK NAEI (2017) scaled at 0.352º × 0.234º and (d) UK NAEI scaled at 0.02º × 0.02º. The NAEI scaled at 0.352º × 0.234º subtracted from the EDGAR emissions (in g s⁻¹) for London is shown in (e) and for the UK in (f). The London region in relation to the UK is shown by the black box in (f).**


# 3 Results

## 3.1 Measurements

The 20-minute averaged $CH_4$ mole fractions and $\delta^{13}CH_4$ values from March 2018 to October 2020 along with the Mace Head background values are shown in Fig. 4a. Mole fractions ranged from 1895 ppb to 3924 ppb in the ICL measurements with a

mean value of $2083 \pm 145$ (1σ) ppb. ICL mole fractions measured during the afternoon (13:00-17:00) were lower on mean, $2028 \pm 73$ (1σ) ppb, and had a lower maximum value, 2477 ppb, showing that higher concentrations are observed during the night-time from the build-up of emissions in the nocturnal boundary layer. The Mace Head background mole fractions ranged from 1907-1973 ppb and had a mean value of $1939 \pm 13$ (1σ) ppb. During the first UK COVID-19 lockdown period (23 March 2020-15 June 2020) we observe more days with higher $CH_4$ mole fractions compared to the preceding months (Fig.

4a). This did not result in a difference between the average mole fractions before and during the UK COVID-19 lockdown period (Fig. 5a).

The $\delta^{13}CH_4$ measurements at ICL are shown in Fig. 4b along with the calculated Mace Head background $\delta^{13}CH_4$ values. The mean $\delta^{13}CH_4$ at ICL for this period is $-47.1 \pm 0.9$ (1σ) ‰ with values ranging from -52.4 ‰ to -42.3 ‰. The afternoon

$\delta^{13}CH_4$ mean was nearly the same, $-47.2 \pm 0.8$ (1σ) ‰. Mace Head background $\delta^{13}CH_4$ averaged $-47.6 \pm 0.2$ (1σ) ‰ and ranged from -48.0 ‰ to -47.4 ‰. Observed $\delta^{13}CH_4$ at ICL was generally higher than $\delta^{13}CH_4$ at Mace Head during 2018, but excursions both higher and lower than the background are seen during 2019-20. We see a mean 0.05 ‰ increase in $\delta^{13}CH_4$ at ICL during the UK COVID-19 lockdown period, but this could be due to seasonal changes rather than anthropogenic influences.

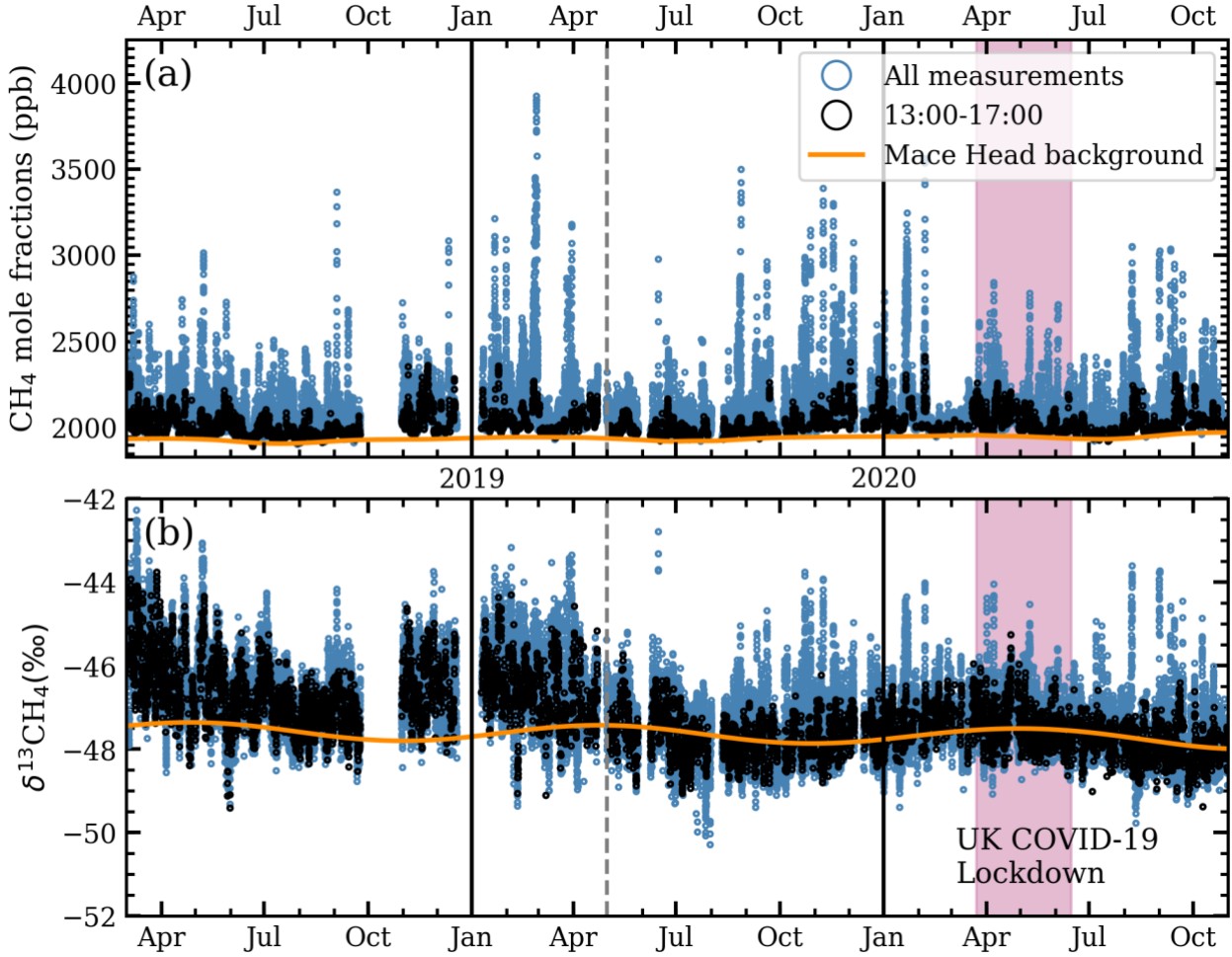

**Figure 4: The 20-minute averaged measured (a) mole fractions and (b) $\delta^{13}CH_4$ values at ICL, along with the daily Mace Head background values from March 2018-October 2020. Afternoon (13:00-17:00) data is shown in black. The period of the first UK national COVID-19 lockdown is denoted by the pink region. The grey dashed line denotes when the standard and target tanks were changed.**

The ICL mole fractions were detrended by fitting a linear polynomial to Mace Head data to find the trend between 2018-2020 with the mole fraction on 1 March 2018 set as the reference point, $t_{ref}$. Detrended mole fractions were binned by month to evaluate seasonal variations (Fig. 5a). A seasonal cycle is observed with a $CH_4$ minimum occurring in July for both ICL and Mace Head measurements. Smaller interquartile ranges and smaller maximum values in the ICL mole fractions are observed in the summer months. Diurnal cycles are observed in the detrended ICL mole fractions with daily minimums between 13:00 and 15:00 (Fig. 6a) with generally smaller mole fractions between April and September. Differences in the diurnal cycles throughout the week vary depending on the time of year. The average nocturnal build-up of $CH_4$ is significantly larger on Monday and Tuesday in the July-August-September (JAS) averaged mole fractions compared to the rest of the week (Fig. 6a),





whereas the October-November-December (OND) averaged mole fractions have relatively similar levels of $CH_4$ nocturnal build-up throughout the week.

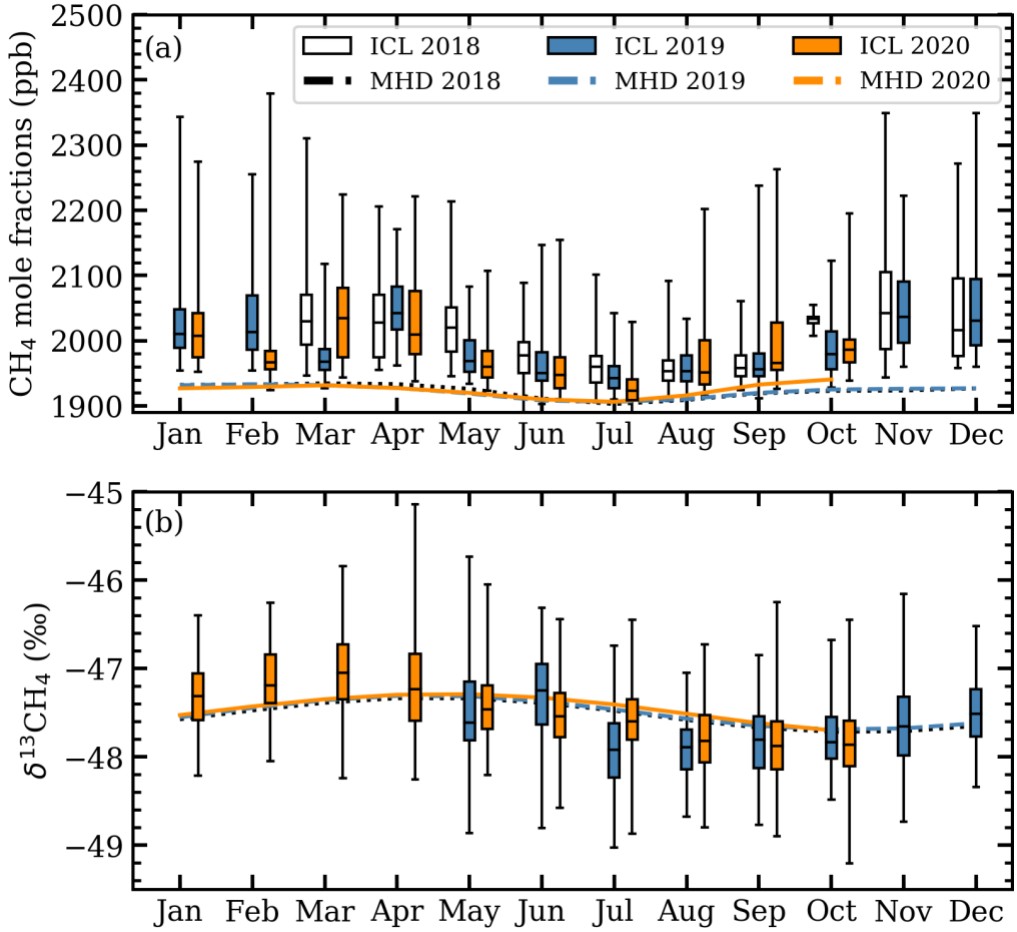

**Figure 5: Seasonal cycles of detrended 20-minute measurements of (a) mole fractions and (b) $\delta^{13}CH_4$ at ICL (box plots) and Mace Head (lines) where values deviate about March 1, 2018 ($t_{ref}$) for mole fractions and about May 1, 2019 ($t_{ref}$) for $\delta^{13}CH_4$.**

We focus on $\delta^{13}CH_4$ measurements from May 2019 onwards in our analysis as the associated measurement uncertainty is smaller (Sect. 2.2.3). Afternoon measurements of $\delta^{13}CH_4$ at ICL were detrended by fitting a linear polynomial to Mace Head background $\delta^{13}CH_4$ from 2018-2020 with $\delta^{13}CH_4$ on 1 May 2019 set as the reference point, $t_{ref}$, (Fig. 5b). ICL median $\delta^{13}CH_4$ between January and March were generally higher than the Mace Head background, and generally lower from July through to September. The $\delta^{13}CH_4$ ICL measurements averaged into hourly intervals tend to exhibit lower $\delta^{13}CH_4$ during the afternoon but no well-defined diurnal or weekly cycle (Fig. 6b).





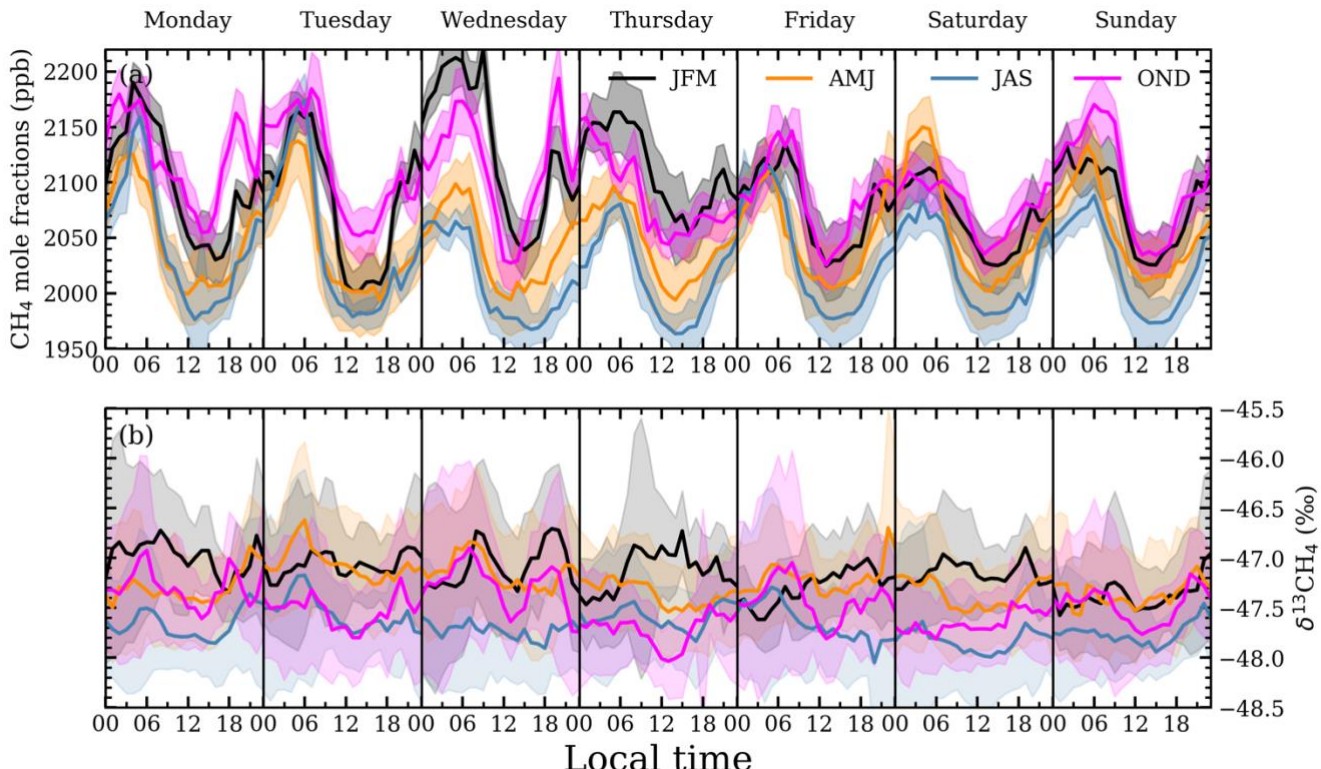

Figure 6: Weekly detrended 20-minute averages of (a) mole fractions and (b) $\delta^{13}CH_4$ at ICL (values normalised to 1 March 2018 and 1 May 2019, $t_{ref}$, respectively). Measurements are grouped by season of year and binned by hour-of-day and day-of-week. The 1σ range is included on both panels.

### 3.1.1 Keeling plot analyses

Three moving time windows of lengths 12 h, 3 days, and 7 days were used in the automated Keeling plot algorithm to find $\delta_s$ values between May 2019 and October 2020 (Fig. 7-8). The calculated $\delta_s$ values may correspond to an individual source sector (Table 3), but they can reflect mixtures of different sources influencing the measured air in each time window, where the $\delta_s$ is a weighted average of the different sources. Isotopic source values lower than -47 ‰ suggest the sources are primarily biogenic (waste and/or agriculture), and $\delta_s$ values higher than -47 ‰ suggest the sources are primarily from gas leaks from the $CH_4$ gas distribution network (i.e. natural gas leaks), where -47 ‰ is the midpoint between the waste and the natural gas $CH_4$ isotopic signatures (Table 3). Isotopic source values are sorted into 5 ‰ bins therefore we use -45 ‰ to distinguish between primarily biogenic and primarily natural gas $CH_4$ sources.

The 12 h moving windows, using measurements from all hours, returned 1046 $\delta_s$ values, of which 24.5 % were ≤-45 ‰. Most of the 12 h pollution events occurred during the nocturnal $CH_4$ build-up and the large number of $\delta_s$ values >-45 ‰ suggests





natural gas sources are primarily driving the nocturnal $CH_4$ build-up around ICL. Natural gas leaks are expected to have a signature of -36±3 ‰ in London (Zazzeri et al., 2017). Uncertainties in $\delta_s$ were 2.8 ‰ in the 12 h windows.

The 3 and 7-day windows using 13:00-17:00 measurements returned 41 and 47 $\delta_s$ values, respectively, and have higher proportions of biogenic influences. In the 3-day windows, 26.3 % of $\delta_s$ values were ≤-45 ‰ and in the 7-day windows, 20.5

355 % of $\delta_s$ values were ≤-45 ‰. Still a majority of pollution events had $\delta_s$ values >-45 ‰, showing that natural gas leaks are the main source of $CH_4$ pollution at ICL sampled in the afternoon and arising from larger-scale regional influences, in addition to the presumably more local sources sampled in the night. Uncertainties in $\delta_s$ were 4.4 ‰ in the 3 and 7-day windows.

The $\delta_s$ values between -30 ‰ and -25 ‰ may arise from a mixture of vehicular and natural gas $CH_4$ but they have mole fraction

peak strengths (Sect. 2.3) smaller than 200 ppb and they comprise less than 5 % of the isotopic source values, indicating $CH_4$ emissions from the nearby roads and power station are small.

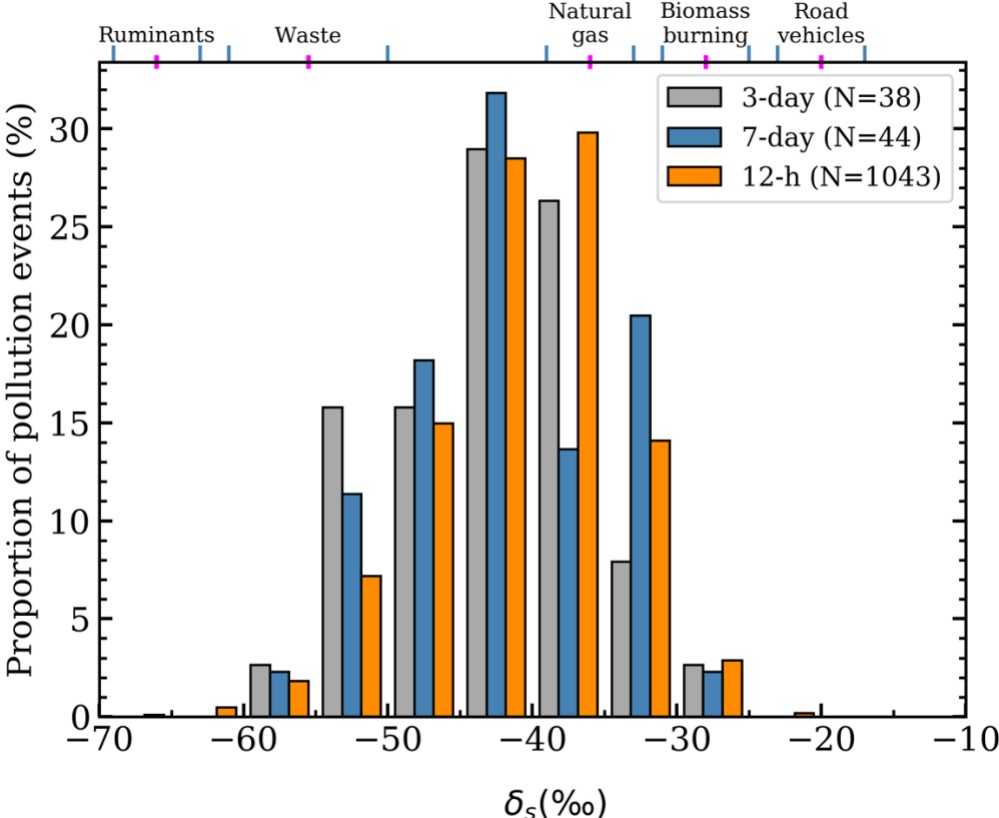

**Figure 7: The distributions of the isotopic source values from Keeling plot analysis. The ranges of different UK isotopic signatures from Zazzeri et al., 2017 are shown at the top for reference.**





We looked for a relationship between wind direction and $\delta_s$ values (Fig. 8) but we do not find any consistent patterns, which

reflects the collocation and heterogeneity of sources in London. Some events with low isotopic signatures and wind direction

in the southerly or south-westerly direction may be influenced by the sewage or landfill sites south or southwest of ICL (Fig.

1). $\delta_s$ values observed during the UK COVID lockdown period were ~2 ‰ higher in the 12 h windows and ~5 ‰ higher in the

3 and 7-day windows compared to the months before and after the lockdown. However, during the UK COVID lockdown

period there was an unusual predominance of easterly winds.

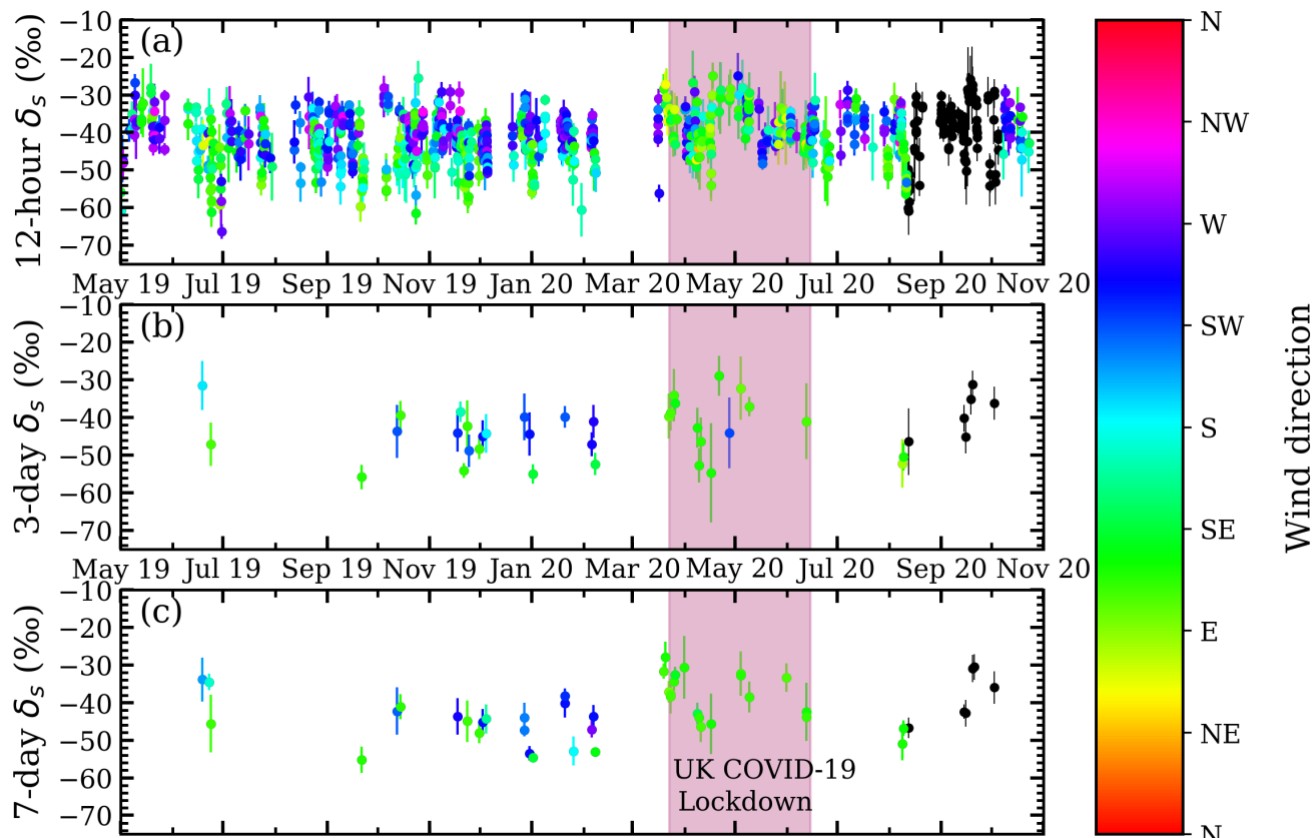

**Figure 8: Time-series of isotopic source values for (a) 12 h; (b) 3-day; (c) 7-day windows. The marker colour denotes the mean wind direction from the start of the window to the peak of the pollution event. Black markers indicate times when wind direction data was not available. The UK COVID-19 lockdown period is shown in pink.**

**3.2 Simulations of methane**

**3.2.1 Simulated CH$_4$ mole fractions**

Simulations of CH$_4$ mole fractions are compared with the observations at ICL in Fig. 9 for 2020 (Fig. S6, S7 for 2018 and

2019) and in Fig. 10 for all years. Simulated CH$_4$ using EDGAR from all hours tends to be higher than the observations, while

13:00-17:00 EDGAR simulations tend to be lower. Simulated CH$_4$ using UK NAEI tends to be lower than the ICL





measurements in both all hours and afternoon data. Higher simulated mole fractions with EDGAR are expected as emissions

in EDGAR are 2.5 times larger than the NAEI emissions for the London area (Table 4).

**Figure 9: Excess simulated and observed mole fractions for 2020 where the Mace Head background has been subtracted from the**
**ICL measurements. (a) shows data from all hours; (b) from between 13:00-17:00.**

The slope of the linear regressions (Fig. 10a-d), the RMSE, and the median simulation-observation differences (Fig. 10e-h)

are used to compare the simulations with the observations. There are small differences between the slope and intercept values

obtained by an ordinary least squares and an orthogonal distance regression. Slopes in the afternoon NAEI simulations are

closer to one than the all data slopes (Fig. 10c-d), whereas the converse is seen for the EDGAR simulations.



Though EDGAR-10km comparisons (Fig. 10b) have slopes closest to one, the EDGAR-10km comparisons also have the largest RMSE (~154 ppb; Table 5), whereas the other simulation-measurement RMSE are between 92 ppb (EDGAR-25km; Table 5) and 114 ppb (NAEI-25km; Table 5).

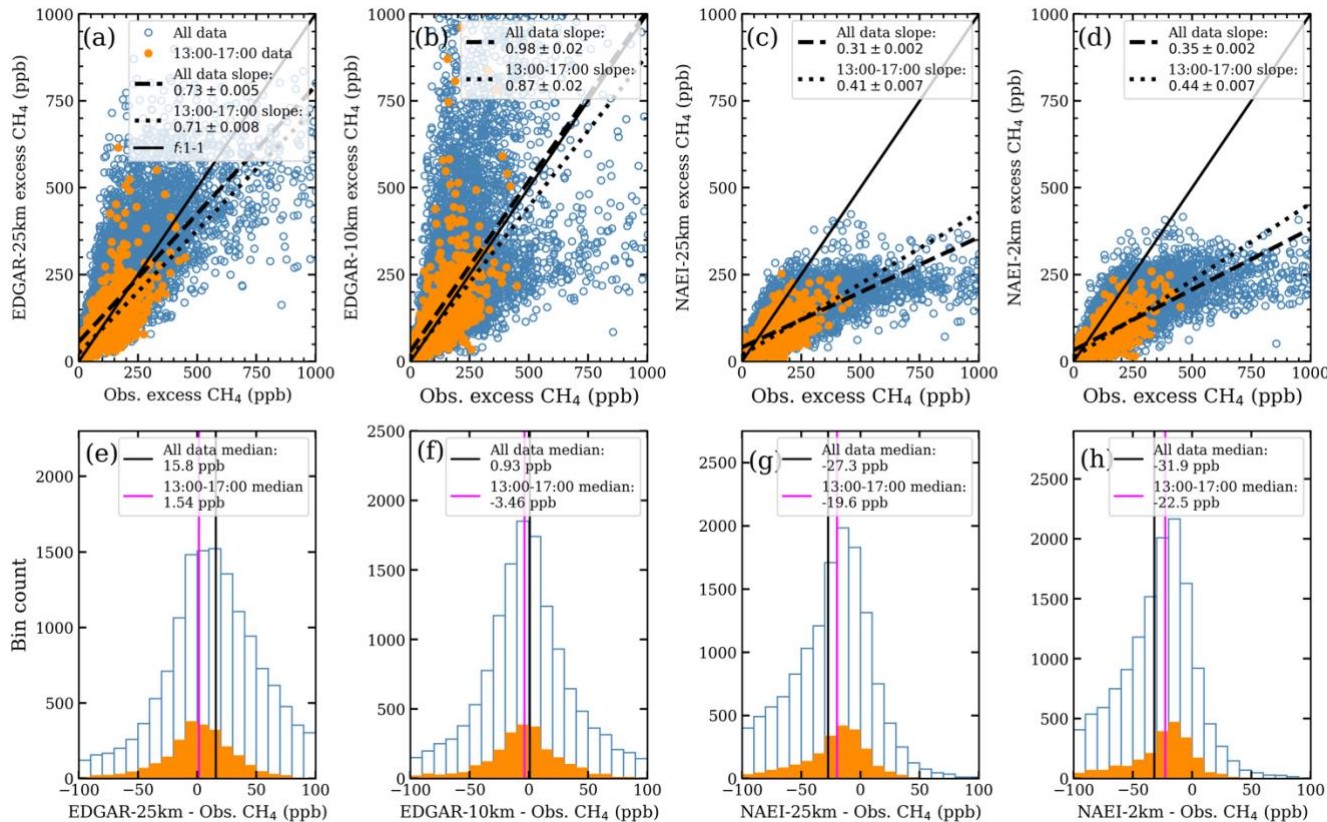

**Figure 10: Simulation-observation comparisons of excess mole fractions using linear regressions (top row) and distributions of the simulation-observation differences (bottom row) for (a, e) EDGAR-25km; (b, f) EDGAR-10km; (c, g) NAEI-25km; (d, h) NAEI-2km from March 2018 to October 2020.**

**Table 5: Simulation-observation RMSE values, scaling factors and correlation coefficients.**

|  | RMSE (all hours) | RMSE (13:00-17:00) | β (all hours) Median (Q1-Q3) | β (13:00-17:00) Median (Q1-Q3) | ρ (all hours) | ρ (13:00-17:00) |
|---|---|---|---|---|---|---|
| EDGAR-25km | 92.1 ppb | 44.5 ppb | 0.84 (0.63-1.14) | 0.97 (0.72-1.29) | 0.78 | 0.74 |
| EDGAR-10km | 154 ppb | 61.9 ppb | 0.99 (0.70-1.36) | 1.07 (0.80-1.46) | 0.66 | 0.66 |
| NAEI-25km | 114 ppb | 52.3 ppb | 1.46 (1.08-1.96) | 1.46 (1.12-1.97) | 0.76 | 0.77 |
| NAEI-2kn | 113 ppb | 53.7 ppb | 1.59 (1.22-2.15) | 1.65 (1.26-2.25) | 0.77 | 0.77 |



Distributions of simulation-observations (Fig. 10e-h) show 13:00-17:00 data have medians closer to zero than data from all hours, except in EDGAR-10km. As previously highlighted, afternoon mole fractions are less sensitive to local emissions and provide a more accurate representation of regional-scale $CH_4$ sources and mole fraction variations. Afternoon weather

conditions tend to be represented better in models as errors in the modelled planetary boundary layer are considered smaller during the afternoon (Brophy et al., 2019; Jeong et al., 2013). EDGAR-10km has the smallest median simulation-measurement differences in all hours and 13:00-17:00 data, where the median difference in the latter is 0.93 ppb. The NAEI-25km and NAEI-2km simulation-measurement distributions have afternoon median values of -19.6 ppb and -22.5 ppb respectively (Fig. 10g-h).


Scaling factors, β, based on the simulation-observation median differences, are calculated by adjusting the simulated values so that they equal the corresponding excess $CH_4$ observation,

$$\beta = \frac{C_{obs}}{C_{sim}}, \quad (3)$$

where $C_{obs}$ are the Mace Head background mole fractions subtracted from the ICL measurements. Background mole fractions

exert a significant leverage on the values of β. We account for this by randomly varying the background mole fractions based on their standard deviations and calculating the β values 150 times.

The median β scaling factors are more similar in the 13:00-17:00 data with EDGAR simulations having scaling factors closer to one (Table 5) suggesting a strong correspondence between the EDGAR emissions and the observations. On average, 13:00-

17:00 NAEI-2km simulations need to be scaled by 1.61 and NAEI-25km by 1.42. NAEI simulations have larger interquartile ranges than the EDGAR simulations, suggesting a higher variability in the NAEI simulated mole fractions.

Increasing the spatial resolution in the simulated mole fractions had a small effect in comparison to the differences between using NAEI and EDGAR emissions for the UK. Our conservative gridding approach (Sect. 2.4.2) ensures emissions across a

region will be the same for all spatial resolutions. Differences will arise as a result of the width of the different back-trajectory plumes and the emissions grid cells they intersect.

### 3.2.2 Simulations of $\delta^{13}CH_4$

Simulated $\delta^{13}CH_4$ values are consistently $^{13}$C-depleted relative to the background in all simulations (Fig. 11, S8), which contrasts with the observations that show $\delta^{13}CH_4$ excursions both above and below the background (Fig. 11). The simulated

range in $\delta^{13}CH_4$ in NAEI-25km and NAEI-2km is only 0.2 ‰, which reflects the strong similarity between the mean isotopic source signature for London of -47.7 ‰ in NAEI (Table 4) and the background $\delta^{13}CH_4$ (-48.0 ‰ to -47.4 ‰). EDGAR-25km and EDGAR-10km also underestimated the variation in $\delta^{13}CH_4$ as isotopically heavy pollution events were missing, even



though the isotopically light spikes are often exaggerated in EDGAR-10km, as was found for the mole fractions. The mean isotopic source signature for London is -53.7 ‰ in EDGAR (Table 4) due to a large proportion of emissions from waste (93 %) and a small proportion from natural gas (3 %). The proportion of emissions from natural gas is higher in NAEI (41 %), but the mean isotopic source signature for London in both NAEI and EDGAR are much lower than the median in the isotopic source signatures calculated in the Keeling plot analysis (-41.6 ‰; Fig. 7)

Simulation-observation comparisons in Fig. 12a-d do not show any correlation between the measurements and the simulations. The simulation-observation difference distributions (Fig. 12e-h) are all negatively skewed and have mean differences ranging from -0.78 ‰, in the NAEI-2km 13:00-17:00 data, to -1.17 ‰, in the EDGAR-25km all data simulations. This indicates the source apportionments in the NAEI and EDGAR inventories have fossil-fractions that are too low, and their sources may be distributed too homogenously.




**Figure 11: Simulated and measured $\delta^{13}CH_4$ values for 2020 using data from (a) all hours and (b) 13:00-17:00. The background values from Mace Head are included for reference.**





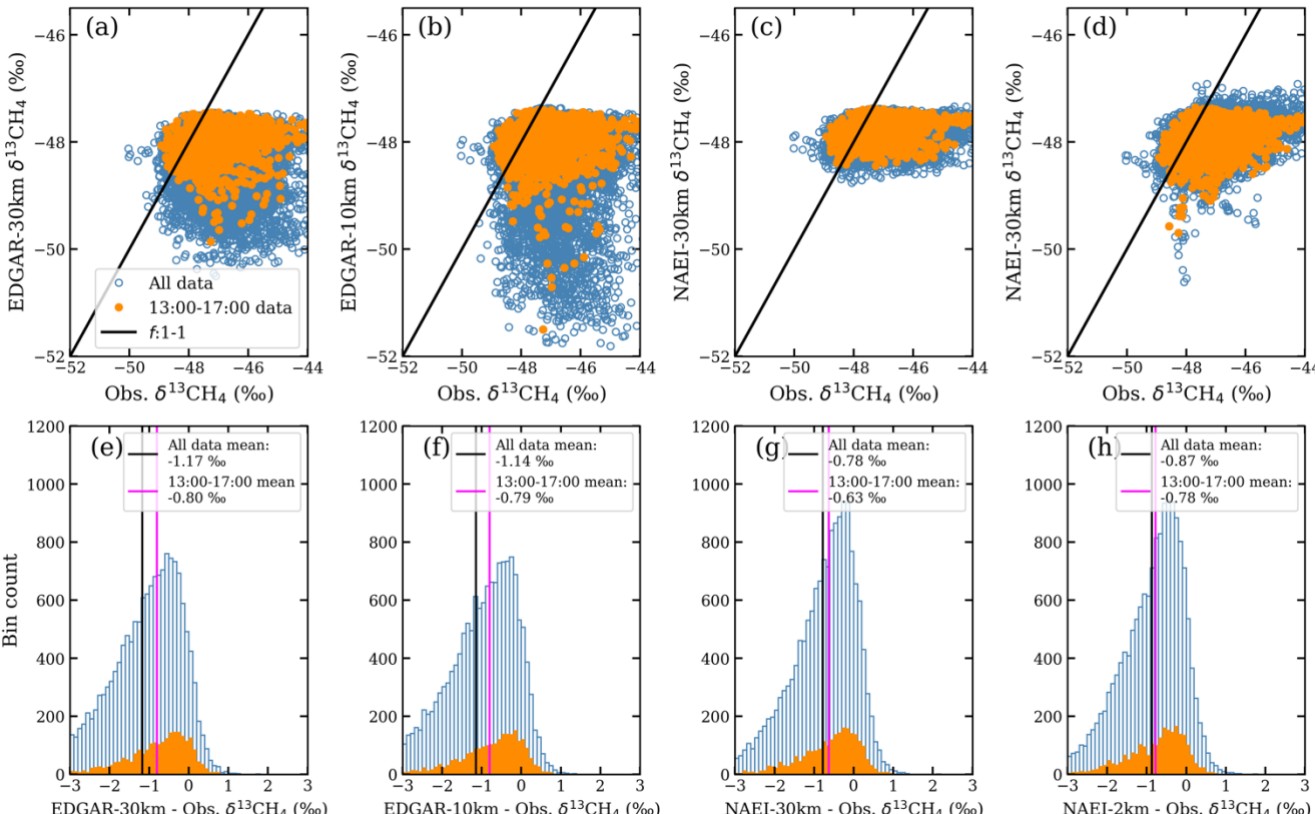

**Figure 12: Simulation-observation comparisons of $\delta^{13}CH_4$ using point-by-point comparisons (top row) and distributions of the simulation-measurement differences (bottom row) for (a, e) EDGAR-25km; (b, f) EDGAR-10km; (c, g) NAEI-25km; (d, h) NAEI-2km.**

To test whether the underestimates in excess $CH_4$ mole fractions and in $\delta^{13}CH_4$ in the NAEI simulations could be explained solely by underestimated emissions from natural gas leak we recalculate $\delta^{13}CH_4$ in NAEI-25km and NAEI-2km by assuming all the missing simulated $CH_4$ is natural gas $CH_4$. Scaling factors for the simulated natural gas mole fractions (Sect. 3.2.3), calculated from the overall $CH_4$ scaling factors (Table 5), are 3.7 for NAEI-25km and 4.1 for NAEI-2km. The recalculated $\delta^{13}CH_4$ shows much smaller excursions below background $\delta^{13}CH_4$ and now some excursions above background $\delta^{13}CH_4$ (Fig. 13), particularly in NAEI-2km where the correlation between observed and simulated $\delta^{13}CH_4$ increased from 0.37 to 0.56. However, it appears that the recalculated $\delta^{13}CH_4$ reflects a rather homogeneous fossil fraction in excess $CH_4$ with an isotopic signature near to background $\delta^{13}CH_4$, which therefore produces very small variations in $\delta^{13}CH_4$ in contrast with the observations. This indicates the locations of natural gas and waste emissions in London are more spatially distinct than in the NAEI inventory.




**Figure 13: Timeseries comparison of simulated (a) NAEI-25km and (c) NAEI-2km $\delta^{13}CH_4$ recalculated by scaling the simulated natural gas mole fractions, along with observations for afternoon hours. Simulation-observation comparisons of $\delta^{13}CH_4$ using linear regressions for (b) NAEI-25km and (d) NAEI-2km for 2020 afternoon hours.**

### 3.2.3 Sectoral source apportionment in the simulations

The mean source apportionment at ICL for each set of simulations are given in Table 6. In all four sets of simulations, $CH_4$ from the waste sector dominated at ICL, accounting for between 30.0 % (NAEI-2km) and 71.1 % (EDGAR-25km) of added $CH_4$ (Table 6). Whilst waste $CH_4$ at ICL was more than three times larger than any other source sector in EDGAR-25km and EDGAR-10km, waste $CH_4$ was lower and more comparable to natural gas $CH_4$ in NAEI-25km and NAEI-2km. Natural gas $CH_4$ at ICL formed the third largest source in the NAEI-25km (20.4 %) and second largest in the NAEI-2km (28.3 %) but it was significantly smaller in EDGAR-25km (6.2 %) and EDGAR-10km (8.1 %). Agricultural sources at ICL accounted for the second largest source in EDGAR-25km (13.8 %), EDGAR-10km (18.8 %), and NAEI-25km (22.2 %).





**Table 6: Mean simulated source apportionment for excess CH₄ at Imperial College London and in the CH₄ emissions for London.**

| Source Sector | Imperial EDGAR-25km (%) | | Imperial EDGAR-10km (%) | | Imperial NAEI-25km (%) | | Imperial NAEI-2km (%) | | Total London: EDGAR (%) | Total London: NAEI (%) |
|---|---|---|---|---|---|---|---|---|---|---|
| | All data | 13:00-17:00 | All data | 13:00-17:00 | All data | 13:00-17:00 | All data | 13:00-17:00 | | |
| Biomass burning | 0.1 | 0.1 | 0.2 | 0.1 | 0.2 | 0.1 | 0.2 | 0.1 | - | - |
| Combustion | 2.5 | 2.3 | 4.1 | 3.4 | 3.5 | 3.2 | 4.4 | 3.7 | 2.9 | 5.5 |
| Natural gas | 6.2 | 7.4 | 8.1 | 8.7 | 20.4 | 17.8 | 28.3 | 22.6 | 3.3 | 41.2 |
| Road vehicles | 0.3 | 0.3 | 0.4 | 0.4 | 0.3 | 0.3 | 0.4 | 0.4 | 0.3 | 0.5 |
| Agricultural | 13.8 | 18.3 | 18.8 | 24.3 | 22.2 | 26.9 | 24.8 | 30.1 | 0.3 | 0.8 |
| Waste | 71.1 | 62.7 | 61.0 | 52.6 | 43.8 | 38.7 | 32.1 | 30.0 | 93.2 | 52.0 |
| Wetlands | 6.0 | 8.9 | 7.4 | 10.5 | 9.6 | 13.0 | 9.8 | 13.1 | - | - |

Higher resolution simulations decreased the proportion of waste sources and increased the proportion of natural gas CH₄ sources. The distribution of emissions in lower resolution simulations are likely to unrealistically smooth the point source emissions from landfills across the London area, increasing the probability of the back-trajectories interacting with emissions from these grid cells. For example, NAEI-2km waste emissions are located towards the outskirts of London (Fig. S9d) but NAEI-25km waste emissions are uniformly distributed across London (Fig. S9c). Similarly, natural gas emissions are located near the centre of London (Fig. S10d) but not uniformly distributed in the coarser resolution emissions due to the absence of natural gas emissions on the outskirts/ outside of London (Fig. S10c).

Simulated CH₄ from biomass burning sources (GFED4) were negligible (<0.2 %; Table 6) in comparison to the contributions from other sources. However, CH₄ from wetlands formed a more significant proportion of added CH₄ (6.0-9.8 %; Table 6), with higher contributions during the summer. A pollution event on 16 August 2019 that had a low isotopic source signature (Sect. 3.1.1) coincided with an 80 ppb simulated wetland mole fraction on the same day.

## 4 Discussion

Continuous measurements of CH₄ mole fractions and $\delta^{13}CH_4$ in central London show, through Keeling plot analyses, a range of different CH₄ sources exist in London. Most isotopic source values are >-45 ‰ indicating a high fossil-fraction of added CH₄ for central London. Comparisons between measurements and the simulated excess mole fractions show a good correspondence between the EDGAR-25km, EDGAR-10km simulations and observations. The NAEI simulations at 2 km and 25 km significantly underestimate the observations, but retain a good correlation. We calculate the NAEI emissions for London



need to be scaled by 1.52 and EDGAR emissions by 0.99, when using the 13:00-17:00 data, which is more representative of the London area and has smaller errors in the modelled boundary layer mole fractions than when night-time data is included. In contrast, we do not observe a correlation between the measured and simulated $\delta^{13}CH_4$ values. Simulations of $\delta^{13}CH_4$ fail to capture any $\delta^{13}CH_4$ excursions above the background as seen in the observations suggesting the NAEI and EDGAR inventories
are underestimating natural gas emissions for the London area.

Under-reported natural gas emissions are reflected in all four $\delta^{13}CH_4$ simulations, where there are few simulated values above the background in contrast to the observations. While the EDGAR-25km and EDGAR-10km mole fraction simulations are most comparable to the observed mole fractions, discrepancies in simulated $\delta^{13}CH_4$ show that the apportionment of sources is
incorrect in EDGAR. Over 90 % of EDGAR $CH_4$ emissions for London are allocated to the waste sector, which would require leak rates in natural gas infrastructure to be very low, in contrast to observations in other cities with older infrastructure (e.g. McKain et al., 2015). The underestimation of mole fractions in the NAEI-25km and NAEI-2km might be accounted for by missing natural gas emissions in the NAEI inventory for London. Scaling the natural gas mole fractions in the NAEI simulations to match the overall excess mole fraction (which increased the natural gas fraction from 22.6 % to 52.1 %)
improved the correspondence between the observations and simulated $\delta^{13}CH_4$ slightly, however, it appears the spatial allocation of waste and natural gas emissions in the inventory is too homogeneous. Overall, it does not seem possible to improve the model-data comparison for both $CH_4$ mole fractions and $\delta^{13}CH_4$ without increasing $CH_4$ emissions from natural gas leaks in the London area in the inventories. More explicit use of $\delta^{13}CH_4$ and $CH_4$ data with high-resolution NAME simulations in an inversion framework including consideration of uncertainties in measured, background and modelled $\delta^{13}CH_4$ and $CH_4$ could
help to specify the fossil fraction in London more precisely.

Both Helfter et al. (2016) and Zazzeri et al. (2017) reported gas leaks are underestimated in London in the emissions inventories as found in other urban areas (Brandt et al., 2014). The median differences between the NAEI simulations and the ICL measurements are not as large as those found by Helfter et al. (2016), however the 2015 NAEI inventory, used by Helfter et
al. (2016), was 46 kt $CH_4$ yr$^{-1}$ larger than the 2017 NAEI inventory across the UK. Our results contrast with Pitt et al. (2019) which found the NAEI inventory was overestimating $CH_4$ emissions for London compared to measurements on a single aircraft flight on 4 March 2016.

The results from these continuous long-term $CH_4$ and $\delta^{13}CH_4$ measurements show that they can be used for effective evaluation
of $CH_4$ emissions from natural gas and waste sources in urban areas. Measurements from a single site would be significantly enhanced by a larger urban network of $CH_4$ and $\delta^{13}CH_4$ measurements encompassing the spatial heterogeneity in different $CH_4$ sources. Measuring from a greater height would also be useful as this would increase the geographical size of the footprint and allow greater mixing of individual sources before measurement.

## 5 Conclusion

This study presents over two years of atmospheric measurements of $CH_4$ mole fractions and $\delta^{13}CH_4$ from Imperial College London. Isotopic source values from Keeling plot analysis revealed a predominance of natural gas $CH_4$ with source values higher than -45 ‰ in ~74-80 % of the afternoon data. In contrast, simulated sectoral contributions using UK NAEI and EDGAR inventories showed the largest fractions from waste sectors, leading to a simulated underestimation of observed $\delta^{13}CH_4$. These results suggest that natural gas leaks in London are under-reported in both inventories, consistent with previous studies in
London and some other global cities.

### Code availability

Python 3 scripts are available upon request.

### Competing interests

The authors declare that they have no conflict of interest.

**Author contributions**

Simulations of $CH_4$ were produced in Python by ES. ES ran the NAME transport model under guidance of AJM and HG for the 10 km and 2 km footprints. AJM provided the 30 km footprints. GZ provided the measurement data and wrote Sect. 2.2. Mole fraction background data from Mace Head was provided by AJM and isotopic values by SEM. HG was the main scientific supervisor and provided guidance on the presentation of results. ES was responsible for the development of the paper, which
forms part of his PhD. All authors provided feedback on the manuscript.

### Acknowledgements

This project was funded by the European Research Council (ERC) under the European Union's Horizon 2020 research and innovation program (grant agreement 679103), the Grantham Institute – Climate Change and the Environment Science and Solutions for a Changing Plant DTP (NE/L002515/1), the Natural Environment Research Council (NERC, UK), the National
Physical Laboratory (NPL, UK). The authors thank the UK Department of Finance for permission to use Northern Ireland UK NAEI data.





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
