# Peer review of "Continuous CH4 and $\delta^{13}$ CH4 measurements in London demonstrate under-reported natural gas leakage"

_Atmospheric Chemistry and Physics, 2021_

## Author Comment (AC1)

**Author responses to referee comments for manuscript ACP-2022-606**

*26 Jan. 2022*

The following document addresses comments from the three referees who reviewed manuscript *ACP-2022-606*. There were no additional comments from the wider scientific community during the discussion period. The original referee comments are shown in black and author responses are shown in blue. Proposed changes to the manuscript are appropriately indicated throughout.

**RC1 referee comments**

On behalf of all the authors, I would like to thank RC1 for taking time to review our manuscript and providing insightful and helpful comments. Your comments and suggestions have been most useful and have helped us improve the manuscript so that it meets the scientific standards of *Atmospheric Chemistry and Physics*. We have addressed all your comments below, with proposed changes to the manuscript stated when relevant.

**General comments**

In the manuscript entitled "Continuous CH4 and δ13CH4 measurements in London demonstrate under-reported natural gas leakage," the authors utilize tower measurements of methane to evaluate the inventory representation of urban methane in London, in both the global EDGAR inventory as well as the national NAEI. The study investigates urban methane emissions which remain, despite the proximity to a large portion of the global population, a poorly characterized part of the methane budget. The finding that methane emissions associated with the natural gas infrastructure are undercounted in inventories is consistent with other studies of urban centers around the global, and points to an area where potential mitigation efforts are tangible, impactful, and requiring of further study.

The methodologies presented are consistent with those established in the literature previously, namely the use of an established atmospheric dispersion model used in conjunction with a gridded emission inventories to generate simulated signals for comparison with observations and isotopic source analysis with Keeling plots. The manuscript is organized in a logical manner and the writing is concise and mostly clear. At times, however, it reads more like a report than a research article. On a several occasions (detailed below in the 'specific comments'), important details or context are missing from the text. With the inclusion of these additional details and discussion, I believe the manuscript meets the threshold for publication and would be of interest to the readers of ACP.

**Specific comments**

Introduction: The authors motivate their work by highlighting previous works investigating urban CH4 emissions, both in London and around the world, as well as other recent studies based on δ13CH4 measurements. There is, however, no statement in the introduction justifying why further measurements are needed. Is it that previous studies have suggested that urban methane is higher than inventoried, but the cause of the discrepancy is not yet know (i.e. need for attribution using isotopic measurements)? A stronger statement of why the presented work is important would aid the reader in understanding how this work adds value to the existing body of work.

We have amended the following paragraphs in Sect. 1 to further justify why additional $CH_4$ measurements are needed for London. We state that whilst previous measurement campaigns allude to emissions inventories underestimating natural gas emissions in London further measurements are needed. The measurement campaign by Helfter et al. (2016) made emission measurements of $CH_4$ but did not make isotopic measurements. The campaign by Zazzeri et al. (2017) evaluated the source apportionment of the inventories through isotopic measurements over a shorter period but did not state whether emissions were missing, misattributed (or a combination of both) in the inventories in comparison to the measurements.

Proposed changes to paragraph 4, beginning on line 45, are shown in bold:

"*Bottom-up $CH_4$ inventories tend to underestimate emissions in comparison to atmospheric measurements in urban regions (Brandt et al., 2014), including in London. Helfter et al. (2016) conducted eddy-covariance measurements from the BT Tower in central London between 2012-2014 and found emissions (72 ± 3 ton km-2 yr-1) were more than double the NAEI inventory values, which was attributed to gas leaks **or effluent $CH_4$ being underestimated** in the inventory (Helfter et al., 2016). Zazzeri et al. (2017) also concluded **from isotopic measurements** that gas leaks were underestimated after finding many large gas leaks in mobile measurement surveys. However, a study using aircraft measurements from a single flight around the London region in 2016 suggested the UK NAEI was overestimating $CH_4$ emissions and they needed to be scaled down by 0.71 (0.66-0.79) to be consistent with the aircraft measurements on this particular day (Pitt et al., 2019). **Additional London measurements are needed to better understand $CH_4$ emissions from different sources and how they compare to updated inventories. In particular, long-term measurements of isotopic composition could provide more robust source attribution than $CH_4$ measurements alone or isotopic measurements from field campaigns.**"*

Proposed changes to paragraph 7 in Sect. 1 beginning on line 74 are shown in bold:

"*Here, we present over two years of continuous measurements of $CH_4$ mole fractions and $\delta^{13}CH_4$ values made from the South Kensington campus of Imperial College London (ICL), in central London; the longest in situ $\delta^{13}CH_4$ measurement campaign reported to date. **The time span of our measurements allowed us to explore relationships between anthropogenic sources at different times of the year, minimise the impact of anomalous pollution events, and assess the impact of the first UK COVID-19 lockdown on $CH_4$ in London.** An automated Keeling plot analysis was created to determine the isotopic source value(s) of individual pollution events.*

***Since previous London $CH_4$ studies there have been revisions to the global and UK national emission inventories. It is important, particularly in urban areas, that updated inventories are evaluated to ensure reported source values are accurate for city-wide mitigation policies to be effective. Unlike some previous London studies,** we compare observations with atmospheric transport model simulations using 2017 UK NAEI and Emissions Database for Global Atmospheric Research (EDGAR) 2012 v4.3.2 (http://edgar.jrc.ec.europa.eu/overview.php; Janssens-Maenhout et al., 2012) bottom-up inventory estimates and their source apportionment for the London region. We used the UK Met Office's Numerical Atmospheric-dispersion Modelling Environment (NAME v7.2; Jones et al., 2007) to transport these emissions under three different spatial resolutions to simulate the excess mole fractions and $\delta^{13}CH_4$ at ICL.*"

Lines 208-210: "To compare the simulated excess $CH_4$ mole fractions to the measurements at ICL, we subtract daily background $CH_4$ mole fractions from the Mace Head Observatory (Arnold et al., 2018; Manning et al., 2011) from the 20-minute averaged measurements at ICL." Is this background methodology consistent with other works? Is that why those references are included? If this is consistent with previous studies, it makes sense to explicitly state that. Is the location of Mace Head Observatory representative as a typical upwind location for the domain? Are there time periods where the $CH_4$ signal at Mace Head Observatory is not representative of the background for the urban domain?

We follow the same methodology as described in Manning et al. (2011), Arnold et al. (2018) to obtain the background values from Mace Head. For times that are not representative of the background (i.e. when air arriving at Mace Head is influenced by local emissions, emissions from Europe or southerly latitudes) these data are excluded, and a background value is obtained by interpolating data.

We address this comment by altering the paragraph beginning on line 208 in the manuscript, with changes shown in bold:

"*Footprints were combined with gridded emissions (Sect. 2.4.2) to simulate $CH_4$ mole fractions above the background mole fractions outside the footprint domain (i.e. excess $CH_4$ mole fractions). To compare the simulated excess $CH_4$ mole fractions to the measurements at ICL, we subtract daily background $CH_4$ mole*

*fractions from the Mace Head Observatory from the 20-minute averaged measurements at ICL. **Daily background CH₄ mole fractions representative of mid-latitude northern hemispheric concentrations are calculated following the methodology presented in Arnold et al. (2018) and Manning et al. (2021).***"

Section 2.4.2: Why use EDGAR v4.3.2 when newer versions are available? Was this the newest version when the work began? If so, would expect anything to change if the newer versions (v5.0, v6) was used instead? This especially relevant because a work is cited (Klausner et al. 2020) that compares their flux measurements to EDGAR v5.0.

When we began this work EDGAR v4.3.2 was the newest version of the global emissions inventory available. Given the large number of simulations to generate and re-analyse this would have taken substantial time to update. We have compared EDGAR v4.3.2 and EDGAR v5.0 and found significant increases in waste emissions between EDGAR v4.3.2 and EDGAR v5.0 in the UK but emissions from the other CH₄ source sectors did not seem to differ between v4.3.2 and v5.0. We have removed the reference to Klausner et al. (2020) that uses EDGAR v5.0 to avoid confusion and what may appear as making a false equivalence between different versions of EDGAR. We added a sentence to the discussion on the difference between EDGAR v4.3.2 and EDGAR v5.0, which would increase the differences between model and observations that we show, because of their larger landfill emissions.

Lines 272-274: "Subtracting the 25 km NAEI emissions from the 25 km EDGAR emissions (Fig. 3e-f) indicates the largest differences between inventories were in cities; London, Birmingham and the Leeds-Sheffield area, which have higher emissions in the EDGAR inventory." What is the takeaway from this statement? That the largest discrepancies in inventory representation of ch4 appear in cities, suggesting that inventory don't capture these emissions well? As written it is not really clear.

We have elaborated the sentence on lines 272-274 to now read:

"*Subtracting the 25 km NAEI emissions from the 25 km EDGAR emissions (Fig. 3e-f) indicates the largest differences between inventories were in cities; London, Birmingham and the Leeds-Sheffield area, which have higher emissions in the EDGAR inventory. **This shows that emissions in urban areas are particularly uncertain and in need of additional constraints.***"

Figure 8: I find this presentation of this data as a time-series difficult to interpret. If the goal is look at the relationship with wind direction and δs, a correlation plot (e.g. wind direction vs. δs) or a polar wind chart would show this more directly.

We have changed Figure 8 to include correlation plots of δs vs. wind direction for the different time windows (see below). We highlight data from during the first UK COVID-19 lockdown by using red markers on the correlation plots. We believe this now clearly illustrates the lack of correlation between isotopic source value and wind direction as well as the higher easterly winds during the first UK COVID-19 lockdown.

Section 3.2.1: I believe the inclusion of nighttime tower observations in this section requires more discussion. As the authors state, the model transport error is smaller in the afternoon. Accordingly, it is not clear from the manuscript as written if the non-afternoon measurements add anything to the findings. Additionally, including nighttime observations is a deviation from several previous tower-based urban studies (including Mckain et al. cited in the introduction), and thus requires more discussion to support the interpretation of this data. I understand that the nighttime observations are used in 12-hour Keeling plot analysis, however, without further information it unclear if in the simulated methane for 'all hours' we are just seeing the influence of higher transport error.

We decided to remove nighttime data results from the model-observation comparisons in the manuscript. As mentioned in the above comment from RC1, the inclusion of nighttime data deviates from previous urban tower-based works. Additionally, the results presented in the manuscript primarily focus on using afternoon data to evaluate inventories as afternoon data minimises local influences and reduces the transport error of the model.

[Figure]

We found larger model-measurement differences and larger interquartile ranges during the night than during the day, which are likely due to the error in the atmospheric transport model, as this is the only changing component in the simulations.

Section 3.2.1: Similar to the previous comment, what does the role of higher transport uncertainty in the non-afternoon hours play in the interpretation of the model-observation mismatch of δ13CH4? In Figure 13 only afternoon hours are shown in the natural gas scaling test. Does the focus on afternoon hours indicate lower confidence in the nighttime simulations?

Following the previous comment we have removed the nighttime hours from the model-data analysis.

Lines 516-521: This paragraph provides some references to other works examining London as points of comparison, but no discussion is included as to why the presented results may or may not differ from these previous works. Without this information it is unclear how the findings presented here fit into the existing body of knowledge for urban methane in London.

We have reworded and expanded the paragraph beginning on line 516 from:

*"Both Helfter et al. (2016) and Zazzeri et al. (2017) reported gas leaks are underestimated in London in the emissions inventories as found in other urban areas (Brandt et al., 2014). The median differences between the NAEI simulations and the ICL measurements are not as large as those found by Helfter et al. (2016), however the 2015 NAEI inventory, used by Helfter et al. (2016), was 46 kt $CH_4$ $yr^{-1}$ larger than the 2017 NAEI inventory*

*across the UK. Our results contrast with Pitt et al. (2019) which found the NAEI inventory was overestimating CH4 emissions for London compared to measurements on a single aircraft flight on 4 March 2016."*

To:

*"Previous ground-based measurement campaigns in London found inventory emissions were underestimated. Helfter et al. (2016) reported mean annual measured emissions of $72\pm3$ t km$^{-2}$ yr$^{-1}$, which was more than double the London inventory estimate. Assuming their measured emissions are representative of the Greater London area, this is approximately equivalent to 0.11 Tg CH$_4$ yr$^{-1}$. This is similar to the EDGAR v4.3.2 (2012) estimate of 0.10 Tg CH$_4$ yr$^{-1}$ for the same London area (Table 4). Simulation-observation comparisons of ICL CH$_4$ mole fractions are in good agreement with the EDGAR emissions estimate suggesting total London CH$_4$ emissions have not significantly changed since the Helfter et al. (2016) measurement campaign. The median differences between the NAEI simulations and ICL measurements are not as large as those found by Helfter et al. (2016) suggesting some improvement in the NAEI emission estimates for London, but with some sources still underestimated.*

*Isotopic measurements of $\delta^{13}CH_4$ by Zazzeri et al. (2017) indicated a predominance of fossil-fuel CH$_4$ in central London that was not seen in the NAEI inventory, which estimated 29 % of London CH$_4$ emissions were natural gas CH$_4$ at that time (compared to 41 % in the current inventory). Whether fossil-fuel CH$_4$ was underreported or misattributed was an open question as Zazzeri et al. (2017) did not use an atmospheric transport model to generate simulations that could be compared with observed concentrations. Our model-data analysis provides evidence that the NAEI inventory does appear to underestimate natural gas leaks, in agreement with Zazzeri et al. 2017's hypothesis."*

**Technical corrections**

Line 184 – The reference to the supplementary material should be to specific section to aid the reader.

Changed "(Supplementary material)" on L185 to "**(Supplementary material: Approach for automated Keeling plot analysis)**"

Figure 2 caption: The version number of EDGAR should be included, especially since newer versions are now available.

Version of EDGAR now included in Fig. 2 caption.

Lines 327-328: "We focus on δ13CH4 measurements from May 2019 onwards in our analysis as the associated measurement uncertainty is smaller (Sect. 2.2.3)." I believe it would really aid the ready to briefly recap why the uncertainty is lower for May 2019 and onward, even just briefly. It is likely the reader will not recall this detail from earlier in the paper.

We have changed line 328 from "*We focus on $\delta^{13}CH_4$ measurements ...*" to "*In our analysis we focus on $\delta^{13}CH_4$ measurements from May 2019 onwards as large unexplained variations in one of the reference tanks before May 2019 result in larger $\delta^{13}CH_4$ uncertainties (Sect. 2.2.3).*"

Lines 413-415: "Background mole fractions exert a significant leverage on the values of β. We account for this by randomly varying the background mole fractions based on their standard deviations and calculating the β values 150 times." It is unclear which standard deviations are being used here. Further clarification is needed.

We have changed lines 415-416 from "*We account for this by randomly varying ...*" to "*We account for this by varying each daily background mole fraction value by randomly sampling from a Gaussian distribution centred on the daily background value and using the daily standard deviation to vary the mole fraction background and calculate the β values 150 times.*"

Line 453: leaks, not leak

Thank you. This has been corrected.

**RC2 referee comments**

On behalf of all the authors, I would like to thank RC2 for reviewing our manuscript. Your comments and suggestions have been insightful and most useful in helping us improve the manuscript so that it meets the scientific standards of *Atmospheric Chemistry and Physics*. We have addressed all your comments below, with proposed changes to the manuscript stated when relevant.

**General comments**

The manuscript of Saboya et al. presents an assessment of bottom-up greenhouse gas (i.e., methane) emissions estimates through direct measurements (mole fractions and carbon isotope data) and simulation methods available for the UK. This is a well written manuscript that contributes to a better understanding of greenhouse gas emission dynamics in urban areas and the usefulness of carbon stable isotope data in this matter. However, I would like the authors to address and discuss on the following points:

**Specific comments**

The authors need to better delineate the overall message from the manuscript. From the first sentence in the abstract (Line 9-10) is not clear what the intention is. Are the authors evaluating the reliability of bottom-up methodologies vs. measured values or the reverse? The authors may need to better define the overall objective to clearly discuss the data.

The objective of this study was to use our measurements of $CH_4$ and $\delta^{13}CH_4$ to evaluate the London emissions and source apportionment reported in the global (EDGAR) and UK national (NAEI) emission inventories.

We have made changes the abstract shown in bold:

*"**Top-down greenhouse gas measurements can be used to independently assess the accuracy of bottom-up emissions estimates**. We report atmospheric methane ($CH_4$) mole fractions and $\delta^{13}CH_4$ measurements from Imperial College London since early 2018 using a Picarro G2201-i analyser. Measurements from March 2018 to October 2020 were compared to simulations of $CH_4$ mole fractions and $\delta^{13}CH_4$ produced using the NAME dispersion model coupled with the UK National Atmospheric Emissions Inventory, UK NAEI, and the global inventory, EDGAR, with model spatial resolutions of ~2 km, ~10 km, and ~25 km. **Simulation-measurement comparisons are used to evaluate the London emissions and source apportionment in the global (EDGAR) and UK national (NAEI) emission inventories.** Observed mole fractions were underestimated by 30-35 % in the NAEI simulations. In contrast, a good correspondence between observations and EDGAR simulations was seen. There was no correlation between the measured and simulated $\delta^{13}CH_4$ values for either NAEI or EDGAR, however, suggesting the inventories' sectoral attributions are incorrect. On average, natural gas sources accounted for 20-28 % of the above background $CH_4$ in the NAEI simulations, and only 6-9 % in the EDGAR simulations. In contrast, nearly 84 % of isotopic source values calculated by Keeling plot analysis (using measurement data from the afternoon) of individual pollution events were higher than -45 ‰, suggesting the primary $CH_4$ sources in London are actually natural gas leaks. The simulation-observation comparison of $CH_4$ mole fractions suggests that total emissions in London are much higher than the NAEI estimate (0.04 Tg $CH_4$ $yr^{-1}$) but close to, or slightly lower than the EDGAR estimate (0.10 Tg $CH_4$ $yr^{-1}$). However, the simulation-observation comparison of $\delta^{13}CH_4$ and the Keeling plot results indicate that emissions due to natural gas leaks in London are being underestimated in both the UK NAEI and EDGAR."*

RC1 also addresses the need for our scientific objective to be better defined. We have made changes to the introduction section that also address this comment.

Lines 45-50: Could the authors better describe how carbon isotope data is usually incorporated into the inventory estimations? Is the isotope data only useful for source identification or they may be used for contribution estimations?

Carbon isotope data (and other atmospheric data) are not usually included in inventory estimates. Inventory emissions are usually formed from statistical datasets. As atmospheric data are not included in inventory

emission estimates these measurements can be used to independently evaluate inventories. We have added the following sentence (changes shown in bold) in the paragraph beginning on line 45:

*"Bottom-up CH₄ inventories tend to underestimate emissions in comparison to atmospheric measurements in urban regions (Brandt et al., 2014), including in London. **Atmospheric measurements can be used to independently evaluate inventory estimates as measurements of the well-mixed atmosphere do not form part of the evidence used to estimate emission inventories.** Helfter et al. (2016)..."*

Lines 151-155: Did the authors apply corrections for potential interferences of hydrocarbons like ethane? What about sulfur from H2S too? Given the vicinity of waste facilities and the influence of local traffic emissions and gas leakage, I consider that more information about the influence of these potential interferences may be needed.

The Picarro G2201-i we used in this study does measure and correct for ethane as part of its measurement of $\delta^{13}CH_4$, but, to our knowledge, it does not do this for H2S. We did not have independent measurements of ethane or H2S and we did not perform any additional corrections, apart from the internal ethane correction built in to the Picarro.

Lines 178-180: Could the authors better explain how the data for the selected time interval (13:00-17:00) were analyzed for the 3-day or 7-day lengths? How were the data aggregated to perform this analysis?

The full methodology for the Keeling plot algorithm is given in the Supplementary Material. A more explicit reference to the supplementary material has been included on line 181.

Lines 364-367: Did the authors explore the influence of the local atmosphere stability (height of the local ABL) rather than the wind direction/speed?

We did not explore the influence of local atmosphere stability as we were interested in seeing if we could determine the likely origin of the sources (determined from Keeling plot analysis) by using our wind measurements to look for correlations with wind direction. Comparing the nighttime vs daytime Keeling plots suggests that we might see lower $\delta^{13}CH_4$ with higher daytime ABLs but we are not sure that an analysis of daytime ABL height vs $\delta^{13}CH_4$ signature would make a useful addition to the paper. We do not measure ABL height so we would have to use the Met Office Unified Model value of the ABL, which we regard to be less reliable at night (see comments above), thus we would have to restrict such an analysis to daytime.

Section 3.2.1 and 3.2.2 (mole fractions and carbon isotope simulations): There are striking differences between the mole fraction biases from EDGAR and NAEI. Mole fraction biases seem to be systematic, but carbon isotope values are rather constant for both EDGAR and NAEI. Could the authors expand on these differences and explaining better the possible factors related with these deviations? This explanation could be inserted in lines 495-500.

We have made changes, shown in bold, in the suggested paragraph in lines 495-500:
*"In contrast, we do not observe a correlation between the measured and simulated $\delta^{13}CH_4$ values. **Unlike the simulated mole fractions, simulated $\delta^{13}CH_4$ values are dependent on the source sector spatial distributions in the emission inventories**. Simulations of $\delta^{13}CH_4$ fail to capture any $\delta^{13}CH_4$ excursions above the background as seen in the observations suggesting the NAEI and EDGAR inventories are underestimating natural gas emissions for the London area."*

Table 3: Could the authors please clarify the UK NAEI SNAP and EDGAR IPCC 1996 sectors nomenclature?

We have expanded the paragraph starting on line 260 to now read:

*"The two inventories use different sectoral definitions. The UK NAEI uses CORINAIR Selected Nomenclature for sources of Air Pollution (SNAP) in which sources are allocated to one of 11 categories. EDGAR follows the 1996 IPCC source sector specification where sources are allocated to one of 7 categories and then further*

*subdivided. For example, emissions from landfills in EDGAR form a subset of waste sector emissions (category number 6) and are thus specified as category 6A (Table 3), whereas in NAEI all waste emissions are aggregated under SNAP 09 (Table 3). Table 3 shows how we aligned the sources between inventories*".

**RC3 referee comments**

On behalf of all the authors, I would like to thank RC3 for taking the time to review our manuscript. Your comments and suggestions have been insightful and most useful in helping us to improve the manuscript so that it meets the scientific standards of *Atmospheric Chemistry and Physics*. We have addressed all your comments below, with proposed changes to the manuscript stated when relevant.

**General comments**

The topic of the manuscript "Continuous CH4 and d13CH4 measurements in London demonstrate under-reported natural gas leakage" by Eric Saboya is very timely provided the need and the political ambition (e.g. EU Methane Strategy as part of European Green Deal) to reduce emissions. In addition, analytics get available to provide sector specific information, applying isotopes or additional tracers.

I agree to the earlier reviews, that the presentation is rather descriptive and would benefit from some more guidance / motivation in all sections. In addition, I have some specific and technical corrections, which should be addressed.

In summary, the manuscript is a valuable contribution to this field of research and the technical quality is very good so worth publishing in ACP after careful revisions.

**Specific comments**

L120: I appreciate the effort the authors invested to assess the quality of their calibration procedure. Nonetheless, best practice should at least be mentioned to guide future studies.

1) Please refer to the respective WMO GGMT guidelines (e.g. https://community.wmo.int/meetings/ggmt-2019). For instance; provide information and uncertainties (CH4, d13C) on the applied standards (air tanks), mention the preference for two-point calibration.

Uncertainties for the primary and working standard tanks have been inserted as an additional paragraph after the first paragraph of Sect. 2.2.3: calibration procedure and measurement uncertainty. "***Primary standards had a $\delta^{13}CH_4$ uncertainty of 0.20 ‰ (JRAS-M16 scale) and a $CH_4$ uncertainty of 0.25 ppb (WMO $CH_4$ X2004A scale). The working standards had uncertainties of 0.2 ppb for $CH_4$ and 0.18 ‰ for $\delta^{13}CH_4$, which are based on the standard deviation of the measurements calibrated against the primary standards. Propagating the error of the primary standard gives a $\delta^{13}CH_4$ uncertainty of 0.27 ‰ for our working standard.***"

We could not perform a two-point calibration because our two working standards (one used as the target tank in our study) do not have enough difference between their $\delta^{13}CH_4$ values (-48.2 ‰ for the standard and -48.5 ‰ for the target tank) and because we assume the working standard being stable over time. However, a two point calibration is preferred to further reduce the uncertainty. We have added the text "***Whilst a two-point calibration yields a smaller uncertainty it could not be performed as the $\delta^{13}CH_4$ values of the two standard tanks (where one is used as the target tank) are too similar, differing by 0.3 ‰, and we assume the working standard is stable over time.***"

In the last GAW report (20th WMO/IAEA Meeting on Carbon Dioxide, Other Greenhouse Gases and Related Measurement Techniques (GGMT-2019)) it is stated:

"*Laboratories starting isotope measurements in atmospheric $CH_4$ may seek to get a suitable range of air mixtures (see round-robin mixtures) in high-pressure cylinders "calibrated" as their highest local reference gases by a laboratory with a well-established referencing history. In the absence of RMs in the form of $CH_4$ (or $CH_4$- mixtures), laboratories may decide to base their calibrations on existing $\delta^{13}CH_4$ or $\delta^2H$-$CH_4$ scale realizations of a well-established laboratory as intermediate solution.*"

We followed the guidelines as the Max Planck Institute who provided our primary standards is one of the well-established laboratories that produce standards for $\delta^{13}CH_4$ in $CH_4$. There are no specific guidelines for calibration procedures of $\delta^{13}CH_4$ reported in this latest report, so each laboratory has to come up with a customised calibration routine. We added the text: "***Specific guidelines for calibration procedures of $\delta^{13}CH_4$***

*are not reported in the latest GAW (20th WMO/IAEA Meeting on Carbon Dioxide, Other Greenhouse Gases and Related Measurement Techniques (GGMT-2019)), so each laboratory has to develop a customised calibration routine."*

2) The applied calibration procedures are somewhat unclear, the term "difference" could be replace by "offset correction". The "d13CH4 ratio calibration", which was finally selected might not be common practice for isotope studies, is there any reference to refer to?

The calibration method was chosen experimentally. The two different approaches were tested on the same measurement set and no difference on the calibrated measurements standard deviations has been observed. Therefore we stick to the default calibration setup of GCWerks, which drift corrects the values using the ratio. We could not find any reference stating that the ratio is better than the offset correction. We have changed "difference" to "**offset correction**" on lines 136 and 139 in the manuscript.

3) The criteria standard deviation of the target tank might not be suitable to decide on the best calibration approach? How about differences between measured and true d13CH4 values, but again, if differences in d13CH4 between calibration and target tank are small, this cannot be tested.

The mean difference between calibrated values and true $\delta^{13}CH_4$ value of the target tank has been measured during the same reference period (May-Nov 2019). However, this difference is negligible for both calibration approaches. Therefore we think that the standard deviation of the target is the best way to assess the two different calibration methods.

L490: Discussion: The authors should discuss the benefit from using additional isotopic (dDCH4) or gaseous tracers (e.g. C2H6).

We have added the following paragraph to the end of the discussion:

"*Measurements of other isotopic tracers, such as deuterium or radiocarbon, or gaseous tracers, such as ethane, would provide additional constraints on the London $CH_4$ source apportionment.*"

**Technical corrections**
L34-36: Please reformulate this sentence to make it better readable.

We have changed the sentence from "*In the 2017 UK NAEI estimates, $CH_4$ from the waste sector is the dominant source in London accounting for 52 % of London's $CH_4$ emissions...*" to "*Across the London area, waste sector $CH_4$ accounts for 52 % of emissions and fossil-fuel $CH_4$ makes up 41 % of emissions (NAEI, 2017).*"

L38-44: This section would fit better after L60?

We have moved this section as recommended.

L94ff: It is not possible to relate the information in the text to Figure 1, e.g. the "~20 small sewage pumping stations and a waste facility south of the site in the Battersea area", some more information on the map or in the legend would be helpful.

We have added numbers to waste facility markers on Fig. 1 (see below) and changed the sentence on line 98 to: "***There are ~20 small sewage pumping stations and a waste facility (marker 3 on Fig. 1) south of the site in the Battersea area (Fig. 1). The precise locations of these small sewage stations are unknown but the approximate area is shown on Fig.1 (Thames Water - personal communication, Oct. 2020).***"

L116: The "Allan precision" and not "variance" should be / and possibly is reported? Please clarify and correct.

We have changed "Allan variance" to "Allan precision" throughout Sect. 2.2.2.

[Figure]

L160: Please state whether there is an effect of H2O on CH4 concentrations? The sentence "A water correction range between 0 % and …" (L 165) should be reformulated.

We added the following sentence to the last paragraph of Sect. 2.2.4 "*We did not find any water interference on the CH₄ mole fraction measurements.*" And reworded sentence on line 166 to "*Two mass flow controllers were used to adjust the flow rates through the bubbler enabling us to calculate the water correction values for water vapour content between 0 % to 2.2 %.*"

L233: Figure 2: the black box could be replace by a different colour to improve visibility.

We explored a range of colours for this box but found black was most appropriate, given the colour palettes available.

L285: On plots a) to e) emissions are provided as log10 values, is it possible to provide "normal" emission values?

There are two or three point sources with high emission values that result in many other sources not being shown on the Figure if we use the inverse-log values. Using logarithm values allows for all sources and their spatial emission distributions to be clearly shown to the readers.

L292: The first section provides information on CH4 mole fractions only, so remove the term "and d13CH4 values".

Have removed "*and δ¹³CH₄ values*" from the sentence.